# Turning toward or away from God: COVID-19 and changes in religious devotion

**Nathan D. Leonhardt**[1☉]*, **Sarah Fahmi**[1☉], **Jennifer E. Stellar**[2], **Emily A. Impett**[1]

**1** Department of Psychology, University of Toronto Mississauga, Mississauga, Ontario, Canada,
**2** Department of Psychology, University of Toronto, Toronto, Ontario, Canada

☉ These authors contributed equally to this work.
* nathan.leonhardt@mail.utoronto.ca

**Data Availability Statement:** Data is available at the Open Science Framework (https://osf.io/a4jcf/).

**Funding:** University of Toronto funds awarded to EAI (206937). The funders had no role in study

## Abstract

Major stressors can influence religiosity, making some people more religious, while making others less religious. In response to the COVID-19 pandemic, we conducted a mixed-method study with a nationally representative sample of religiously affiliated American adults ($N$ = 685) to assess group differences between those who decreased, stayed the same, or increased in their religious devotion. In quantitative analyses we evaluated differences on sociodemographic variables, religious behaviors, individual differences, prosocial emotions, well-being, and COVID-19 attitudes and behaviors. Of most note, those who changed (i.e., increased or decreased) in religious devotion were more likely than those with no change in devotion to experience high levels of stress and threat related to COVID-19, but only those who increased in religious devotion had the highest dispositional prosocial emotions (i.e., gratitude and awe). Further, those who changed in religious devotion were more likely to report searching for meaning than those with no change, but only those who increased were more likely to report actual presence of meaning. Qualitative analyses revealed that those who increased in religious devotion reported increasing personal worship, the need for a higher power, and uncertainty in life as reasons for their increase in religious devotion; those who decreased reported being unable to communally worship, a lack of commitment or priority, and challenges making it hard to believe in God as reasons for their decrease in religious devotion. The findings help identify how COVID-19 has affected religious devotion, and how religion might be used as a coping mechanism during a major life stressor.

## Introduction

Religion can offer an important source of comfort during major life stressors by providing a framework through which an individual can interpret life events and regulate their emotions [1]. As a result, religious devotion often increases in the wake of major life stressors [2], including natural disasters such as earthquakes [3] and hurricanes [4]. However, major stressors do not increase religious devotion for everyone. Some people may interpret negative life events as a punishment from God and feel they are unable to rely on God as a source of help [2], possibly

design, data collection and analysis, decision to publish, or preparation of the manuscript.

**Competing interests:** The authors have declared that no competing interests exist.

even feeling angry toward God [5]. These individuals may then experience spiritual tensions, which can lead to feeling abandoned by or having an insecure relationship with God [2] as well as doubting their religion [6].

The COVID-19 pandemic presents a unique opportunity to more comprehensively understand how a large population might increase or decrease their religious devotion in response to a major, societal-level, and chronic stressful experience. In March 2020, immediately after COVID-19 was declared an international pandemic, an analysis across 95 countries revealed a 300% increase in Google searches for the term "prayer" [7]; another survey conducted at the same time found that more than half of Americans had prayed in the hopes of ending the pandemic [8], and additional surveys have found that people were reading religious texts or carrying religious items for protection [9], and that many increased in spiritual fortitude [10], 2021. Some research has suggested that these efforts have been effective, as positive religious coping has been connected with less stress [11], religion in general has been connected to positive affect [12], and spirituality has been connected to reduced hopelessness [13]. On the other side, a small, but notable portion of people seem to have experienced reduced religious faith in response to the COVID-19 pandemic. A recent sample of American adults found that 2%, which would be roughly 6.5 million Americans, reported that COVID-19 has made them less religious [14]. Additionally, the onset of COVID-19 resulted in less of a faith mindset and more of a science mindset in the early months of the pandemic [15] and research has shown that negative religious coping has a stronger positive association with COVID-19 anxiety than positive religious coping has negative association with COVID-19 anxiety [16].

Despite a growing body of research highlighting changes in religious devotion occurring along with major stressors, along with characteristics connected with those changes, much remains to be learned about characteristics that specifically differentiate those who increase, decrease, or remain stable in their religious devotion during this tumultuous period. Understanding what factors distinguish people who increase versus decrease in religious devotion may help us understand how and why some people benefit from turning to religion in times of stress, versus why some become more skeptical of their faith [17]. In the present investigation, we collected a nationally representative sample of individuals in the U.S. to identify what differentiated those who increased, decreased, and were stable in religious devotion in the face of COVID-19. Our approach was mixed-methods and included both qualitative and quantitative analyses. We used quantitative analyses to examine group differences for of those who decreased, stayed the same, or increased in religious in response to the pandemic. We also used qualitative analyses that allowed people to communicate how and why their religious devotion had changed, if it had, potentially providing greater insight into explanation for the differences in characteristics. To understand what differentiated those who reported increasing, decreasing, or staying the same in religious devotion, we explored several, theoretically-motivated groups of variables, including sociodemographics (e.g., gender, ethnicity), religious practices, well-being (i.e., life satisfaction, meaning in life), prosocial emotions (e.g., gratitude, awe), and COVID-19 attitudes and behaviors (e.g., stress from COVID-19, social distancing behaviors).

## Sociodemographic variables

One group of variables that we explored might relate to change and stability in religious devotion include the sociodemographic variables of gender, ethnicity, socioeconomic status, education, political orientation, work hours, number of kids and relationship status. For gender, Maynard and colleagues [18] found women to be more likely than men to turn to religion as means of coping as they are more likely to surrender to wait for God to take control, suggesting

women may be more likely to turn to religion in times of stress. For ethnicity, Black communities have been disproportionately hurt by COVID-19 [19] and some past research evidence has shown that Black Americans facing anxiety were more likely to turn to religion compared to White Americans [20]. The disproportionate stress response of those in lower socioeconomic statuses being affected by COVID-19 through overcrowded accommodation, poor housing conditions, and limited access to personal outdoor space [21] could result in searching for a coping mechanism, with religion being a viable option. Concerning education, since individuals who are more educated tend to be less religious [22], they may be less likely to turn to religion and more likely turn to alternative sources as a way to cope with stress, such as scientific development and news on the vaccine. For political orientation, recent research suggests that those who are more politically conservative are less likely to view themselves as personally vulnerable to COVID-19 and less likely to view the virus as being severe [23]; by viewing the virus as less threatening they may be less likely to seek a coping mechanism and thus not feel a need to adapt religiously. Unemployment or reduced work hours may bring added stress similar to the stress brought on by lower socioeconomic status; the time scarcity hypothesis suggests that as people devote more time to their jobs, the less time they have for religious involvement [24–26], suggesting people might find more time for religious involvement and have a chance to increase their religious devotion. Religious individuals have more kids than those who are not [27], meaning that they may deal with added stress from children being home [28]; some research suggests that religion can help reduce parenting stress and increase satisfaction [29], but we do not know whether the stress that comes from parenting may be more likely to influence someone to increase or decrease in religious devotion. Finally, with regards to relationship status, research has shown that religious groups hold the highest number of married couples compared to atheist or agnostic groups that have higher numbers of people who have never been married [30]. Considering that those who are not in a relationship are more likely to experience worse mental health from COVID-19 [31], those outside of the security of a committed, marital relationship may be more likely to experience stress from not being connected to people. Religion can potentially be a source of comfort and intimacy [32] that may be helpful in the face of being more isolated, but we do not know whether someone is likely to increase or decrease religious devotion based on the stress from being more socially isolated.

## Religion variables

A second group of variables that we explored might relate to change and stability in religious devotion include the specific religious behaviors of attending worship services, talking about religion, praying, and reading sacred texts, as well as religious coping and gratitude to God. The way that COVID-19 regulations affect the environment suggests that an overarching increase or decrease in religious devotion may differentially manifest in several religious behaviors. For example, COVID-19 social distancing policies and regulations surrounding gathering for religious services could make it particularly challenging to engage in more extrinsic religious behaviors such as attending church and having religious talks with others [33], unless people are able to maintain a sense of belongingness in their community [34]. On the other hand, the evidence so far suggests that those who report an increase in religious devotion seem particularly likely to increase in more private and internal religious behaviors, such as prayer and the reading of sacred texts [7, 35]. Turning to a higher power in search of meaning, control, comfort, or other religious coping factors may be particularly manifest in private religious behaviors such as the frequency of prayer and reading sacred texts, a finding that would be consistent with other research showing intrinsic religiosity tends to increase following a crisis [36], and that ritualistic behavior helps reduce anxiety [37]. In addition, higher

levels of religious coping [32] and gratitude to God [38] can also be expected when turning to a higher power. There may also be some complications when stress or anxiety is concerned, as one study highlighted that those who began the pandemic as strong believers and had high anxiety about COVID-19 increased their religious beliefs, whereas those with less believe and anxiety about COVID-19 became more skeptical of their beliefs [17].

## Individual differences

A third group of variables that we explored might relate to change and stability in religious devotion include individual differences in loneliness, Big-5 personality traits, and need for closure. Religion may provide a way for people to cope with loneliness, as some research shows that loneliness tends to decrease when people are religious [39–41], but one challenge to this finding, in light of COVID-19, is that the alleviation of loneliness was explained by the fact that religiosity encourages people to attend worship services [39]. In terms of personality traits, one study showed that individuals with higher levels of extraversion, conscientiousness, and openness to experience fared better affectively in response to a stressor, whereas those higher in neuroticism and agreeableness fared worse [42]. These personality traits that make it more likely that people fare worse affectively in response to a stressor may also make it likely that people might be more likely to change their religious devotion in response to a stressor, although we do not know whether the response to stress would lead to an increase or decrease in religious devotion. An additional consideration for extraversion is that extraverted people may view religion more in terms of a social aspect for worship service attendance, and with those social opportunities limited, they may report a decrease in religious devotion. Finally, those with a higher need for closure (i.e., motivation to have answers in uncertainty) likely had a particularly challenging time adjusting to the uncertainty brought on during the early days of the pandemic. As religion can provide a framework to make meaning of important life events, indeed, several studies have shown that religiosity can reduce feelings of uncertainty [43, 44], and individuals with higher need for closure may turn to religion as a way to make sense of the tenuous life circumstances brought on by the pandemic. In fact, one study specifically showed that psalm recitation was used to cope with uncertainty during a war [45].

## Prosocial emotions

A fourth group of variables that we explored might relate to change and stability in religious devotion include prosocial emotions that help individuals transcend self-interest such as gratitude and awe [46]. These emotions have a rich history in religion, with a robust literature demonstrating links between these prosocial emotions and religiosity [38, 47–50]. The tendency to experience these emotions may be an important element of increased devotion during times of stress. For instance, turning to God can help people realize the blessings they have during these negative times, which can increase a sense of gratitude [51]. Similarly, the tendency to experience awe may promote devotion through increasing engagement with religious rituals, which brings a sense of connection to something greater than oneself. Alternatively, turning to God could trigger engagement in religious rituals that can instill awe [50]. Regardless of directionality between these variables, we expected that those who reported an increase in religious devotion would also report greater awe and gratitude.

## Well-being

A fifth group of variables that we explored might relate to change and stability in religious devotion include indicators of well-being such as life satisfaction, presence of meaning in life, and search for meaning in life. Changes in religious devotion are associated with changes in

key indicators of well-being such as life satisfaction [52] and meaning (both the search for and presence of meaning). For instance, in the face of stress, researchers have observed that those who cope by turning to God have more meaning present in their lives, whereas those who turned away from God tended to search for meaning [53]. Therefore, those who report an increase in religious devotion during the COVID-19 pandemic may also report higher presence of meaning and higher life satisfaction, as religion can help people make sense of the purpose of life [32]. In contrast, those who report a decrease in religious devotion may report a greater search for meaning and have lower life satisfaction, due to a destabilization of what was once likely a source of meaning making in their worldview.

## COVID-19 specific attitudes and behaviors

A sixth and final group of variables that we explored might relate to change and stability in religious devotion include COVID-19 specific attitudes and behaviors including level of stress, COVID-19 as a threat to health, finances, and the economy, beliefs about social distancing (e.g., morally wrong to violate social distancing policies), belief that God would protect them from COVID-19, social distancing behaviors, COVID-19 specific prosocial behaviors, and motivations for social distancing. Since we collected data toward the beginning of the pandemic, the measures we included to capture COVID-19 specific attitudes and behaviors were highly exploratory, so we did not have specific predictions. However, we did reason that some differences might emerge. In general, it seems that attitudes and behaviors that suggest highlighting COVID-19 as a stressor and threat may result in change in religious devotion. For example, if COVID-19 is not viewed as threatening, an individual may be unlikely to make any changes from COVID-19, religious or otherwise. By extension, those who generally view COVID-19 as more threatening may be more likely to take social distancing behaviors more seriously. Some might suggest an alternative line of thought, as some religious individuals have believed that God would protect them from the virus, and that they did not need to socially distance, wear masks, or even use hand sanitizer [54]. Researchers have found that highly religious Americans were less concerned about COVID-19, less likely to follow restrictions implemented by public health, less likely to socially distance, and less likely to experience stress compared to non-religious individuals [55]. Additionally, a study highlighted that highly religious people were more likely to engage in unreasonable behavior like hoarding toilet paper, but there were no differences in reasonable behavior like avoiding physical contact or frequent handwashing [56]. With most of these studies focusing on overall religiosity, much remains to be learned about how various COVID-19 attitudes and behaviors might be differentiated by changes in religious devotion.

## Current study

Altogether, we explored the role of several groups of factors including sociodemographic variables, religion variables, individual differences, prosocial emotions, well-being, and COVID-19 specific attitudes and behaviors in exploring change and stability in religious devotion in respond to the COVID-19 pandemic. We collected data between May 27–31, 2020, prior to our preregistration. After cleaning the data and running basic frequencies, we uploaded a preregistration detailing our research question and our qualitative data analysis plan. Once we completed the qualitative data coding and organizing of themes, we used a combination of theory and the information gleaned from the qualitative data to preregister our quantitative analysis. Both preregistered analyses are available on the Open Science Framework, as well as data, syntax for the quantitative analysis, and codebook showing all variables from the dataset whether used or unused in our analysis.

In our quantitative analyses, we presented participants with a variety of established scales to identify the characteristics associated with increases, decreases, or stability in religious devotion during COVID-19. Then, we allowed participants to express in their own words the reasons they increased, stayed the same, or decreased in religious devotion and coded these data in our qualitative analyses. The combination of quantitative and qualitative approaches marks a number of distinct advantages in this work. Guided by a bottom-up approach, qualitative approaches give voice to how participants think and feel about the role of COVID in their religious lives. Quantitative findings allow us to test a variety of differentiating variables in a top-down manner. Insofar as findings can be synthesized, qualitative findings further explain and add depth of understanding to the quantitative findings, potentially providing some explanations to description from quantitative findings, while quantitative findings can help us have a better sense of what qualitative findings might be more generalizable [57, 58]. Despite preregistering the qualitative analyses before the quantitative analyses, we have elected to report the quantitative analysis before the qualitative analysis, as we sought to provide readers first with a broader overview of which factors are associated with change and stability in religious devotion before diving more deeply into potential reasons behind the quantitative findings.

## Method

### Participants

We recruited 1,021 participants from the United States online through the Prolific Academic platform, a crowdsourcing website that produces high quality data [59]. Participants were paid the equivalent of $2.50 USD. Prolific Academic helps researchers, through filtered questions, reach their targeted population and post their study to eligible participants. We used quota sampling on Prolific to gather a nationally representative sample in terms of gender, ethnicity, and age. Of our 1,021 initial participants, we removed those who reported being agnostic, atheist or having no religious affiliation (n = 336) for a final sample of 685 participants. We removed these participants because those without an institutional religious affiliation suggested that some of the questions were not applicable to them in their qualitative responses. In addition, we found that no change in religious affiliation was reported among 90% of agnostics, 92% of atheists and 88% of those who did not report having a formal religious affiliation. On average 6% of each of these groups reported an increase in religious devotion and 3% reported a decrease, a substantially different pattern than those with a religious affiliation. In the final sample, 47% were men and 53% women; 71% were White, 14% Black, 7% Asian, 6% Hispanic, 1% Middle Eastern, and 1% Native American; 43% were Protestant, 33% Catholic, 5% Jewish, 3% Islamic, 3% Buddhist, and 13% reported an "Other" religion that was often a more specific denomination of Christianity. Age ranged between 18–78 with a mean age of 24.0 (*SD* = 15.1). The study was approved by the institutional review board at the University of Toronto. All participants gave written informed consent.

### Quantitative procedure

Participants took part in the study from their own personal computers on the platform Qualtrics. Before completing the survey, participants completed a consent form where they were informed that they were allowed to withdraw from the study or refrain from answering any questions. First participants filled out measures of their religiosity, religious practices, perceived severity of COVID-19, prosocial emotions, and well-being. At the end of the study, they were debriefed.

## Measures

**Sociodemographics.** We asked participants about their gender ("Man", "Woman" or "Other or prefer not to say"), ethnicity ("White or European American" "Black or African American", "Asian or Asian-American", "Hispanic or Latino-American", "Middle Eastern", "Native American" or "other") and education (1 = Less than High School; 2 = High School; 3 = Some College, No Degree; 4 = 2-year Degree; 5 = 4-year degree; 6 = Master's Degree; 7 = Doctorate). We also included measures regarding work hours, social class (1 = Lower class; 2 = Lower middle class; 3 = Middle Class; 4 = Upper middle class; 5 = Upper class) income level (0 = None; 11 = over $300,000 yearly gross income) and political orientation (1 = Liberal to 10 = Conservative). Finally, we asked participants to report on their relationship status (Single; In a romantic relationship; Cohabiting; Married) and number of children.

**Religion.** To assess participants' religious devotion change, participants reported on the degree of stability versus change in their religious devotion on a 5-point scale (1 = Decreased strongly, 2 = Decreased slightly, 3 = Stayed the same, 4 = Increased slightly, 5 = Increased strongly). To ensure a large enough number of participants to fill categories in order to make meaningful comparisons, "Decreased Strongly" and "Decreased Slightly" were collapsed into "Decreased" and "Increased Slightly" and "Increased Strongly" were collapsed into "Increased." Then, they answered an open-ended question (in 1–2 sentences) in which they were asked to report on why their religious devotion had changed (or stayed the same) since the onset of COVID-19. We also included items assessing their frequency of engagement in various in religious behaviors both before and after the onset of COVID-19, including ("Talking about religious and/or spiritual topics in a typical week", "Praying in a typical week", "Reading spiritual or religious text in a typical week" and "Attending religious service in a typical month") assessed on a frequency scale (from 0 to 7+ times). Religious coping during COVID-19 was assessed with a 3-item measure of benevolent religious reappraisal (e.g., I see my situation as part of God's plan"; [60]; Cronbach's $\alpha$ = .87) on a 7-point scale (1 = Strongly Disagree to 7 = Strongly Agree). Participants also indicated their level of gratitude to God with ten items (e.g., "My life is filled with God's grace"; J. Langston, personal communication, January 23, 2020; $\alpha$ = .95) on a 9-point scale (1 = Strongly Disagree to 9 = Strongly Agree).

**Individual differences.** We assessed individual differences in need for closure with 15 items (e.g., "I don't like situations that are uncertain") measured on a 7-point scale (1 = Strongly disagree to 7 = Strongly

Agree; e.g. "I don't like situations that are uncertain"; [61]; $\alpha$ = .86). Second, we measured loneliness with five items (e.g., "I feel left out") on a 4-point scale (1 = Never to 4 = Often; [62]; $\alpha$ = .91). Third, we assessed individual differences in Big 5 personality traits of extraversion (e.g., "I see myself as extraverted, enthusiastic"; $\alpha$ = .68), agreeableness (e.g., "I see myself as sympathetic, warm"; $\alpha$ = .33), conscientiousness (e.g., "I see myself as dependable, self-disciplined; $\alpha$ = .65), neuroticism (e.g., "I see myself as anxious, easily upset"; a = .74), and openness (e.g., "I see myself open to new experience, complex"; $\alpha$ = .42) with the Ten Item Personality Inventory (TIPI), with items answered on a 7-point scale (1 = Strongly Disagree to 7 = Strongly Agree) [63].

**Prosocial emotion.** We measured prosocial emotions with assessments of gratitude and awe. Participant answered six items about general gratitude (e.g., "I have so much in life to be thankful for") through the Gratitude Questionnaire (GQ-6; [64]), assessed on a 7-point scale (1 = Strongly agree to 7 = Strongly disagree; $\alpha$ = .82). Participants also completed six items about awe (e.g., "I often feel awe"; [65]; $\alpha$ = .84), with items rated on a 7-point scale (1 = Strongly agree to 7 = Strongly disagree).

**Well-being.** We assessed well-being with measures of satisfaction with life and meaning in life. Satisfaction with life was measured with five questions (e.g., "In most ways my life is

close to ideal") on a 5-point scale (1 = Strongly Disagree to 7 = Strongly Agree; [66]; α = 0.91). We measured meaning in life with the Meaning in Life Questionnaire (MLQ; [67], 2006), which contained two 5-item subscales including search for meaning (e.g., "I am always looking to find my life's purpose; α = .95) and presence of meaning (e.g., "I understand my life's meaning"; α = .91), with items assessed on a 7-point scale (1 = Absolutely untrue to 7 = Absolutely true).

**COVID-19 attitudes and behaviors.** We included several measures to assess participants' attitudes and behaviors towards COVID-19. First, we developed a set of 10 items for the purposes of this research to measure participants' beliefs about the extent COVID-19 is a threat (e.g., "How serious of a threat do you believe COVID-19 is to your personal health?") rated on a 7-point scale (1 = Not at All to 7 = Very Much). Through exploratory factor analysis we found that the COVID-19 belief items best loaded onto four factors: regulations, health, personal finance, and world finance. S1 Table shows that all items clearly loaded onto each of their respective factors (all above .74) and had limited cross-loading with any other factor (highest cross-load was .16). All four factors included an eigenvalue above one, amounting to a cumulative variance of 76.44% (see S1 Fig for a scree plot). Two items indicated higher agreement with ending COVID-19 regulations (e.g., "I believe it is time to end the sheltering at home"; α = .88). Four items focused on health concerns from COVID-19 (e.g., "How serious of a threat do you believe COVID-19 is to your personal health"; α = .92). Two items focused on personal financial concerns (e.g., "How serious of a threat do you believe COVID-19 is to your financial situation"; α = .90) and two items focused on wider economic concerns (e.g., "How serious of a threat do you believe COVID-19 is to the world economy?"; α = .93). We also had three individual items that failed to clearly load onto any factor, which we assessed as unique constructs, "It is morally wrong to violate social distancing policies right now," "How much stress do you feel from COVID-19," and "I believe that God will protect me from COVID-19" all on a seven-point scale (1 = *not at all* to 7 = *very much*).

We also measured participants' efforts to social distance with 10 items (e.g., "Avoided grocery stores at peak times" developed by [68] rated on a 7-point Likert scale (1 = Not at all to 7 = Completely; α = .81).

Additionally, we conducted a factor analysis on motivations to social distance, which revealed that five items held well together (e.g., "To avoid getting others sick"; α = .85). We called this factor *Avoid Sickness*. Two additional items loaded onto a different factor, "Because the government has required it" and "So I don't get in trouble" and had a moderately high correlation with each other ($r = .47$, $p < .001$) and decent reliability for two items (α = .65). We called this factor *Avoid Punishment*. The scale ranged from 1 = Not at All to 7 = Very Much. S2 Table shows that all items clearly loaded onto each of their respective factors (all above .61) and had limited cross-loading with any other factor (highest cross-load was .09). Both factors included an eigenvalue above one, amounting to a cumulative variance of 66.86% (see S2 Fig for a scree plot).

Finally, we measured prosocial behavior during COVID-19 with four items (1 = Not at all to 7 = Very much) created specifically for this study that loaded onto one factor (e.g., "I have donated money in the struggle against COVID-19"; α = .71).

## Qualitative procedure

We divided the religiously affiliated participants into three groups based on their responses to the question about change in religious devotion (same groups as the quantitative portion). For each of these three groups, we conducted two-step qualitative analyses on their explanation for their change or lack of change in religious devotion. Participants answered an open-ended

question (in 1–2 sentences) in which they were asked to report on why their religious devotion had changed, or stayed the same, since the onset of COVID-19. Utilizing in vivo coding [69], we annotated the exact word or short phrase reported by each participant regarding why their religious devotion had changed or did not change. Two coders read each of the responses and annotated any words or phrases they believed provided insight into our overarching question. Having two coders helped to ensure maximum coverage of annotations that could be grouped into themes.

Second, we organized the annotations into broader themes. The two coders first consolidated their annotations into a single working document. They then discussed each annotation and worked together to organize all of the annotations into themes. Coders came to agreement on all codes through discussion. They then organized themes based on the annotations and all authors were consulted and agreed upon the final themes that were reported. Through this process we used a combination of thematic analysis (identifying, analyzing, and interpreting patterns of meaning within qualitative data; [70]) and focused coding (identifying recurrent patterns and multiple layers of meaning; [69]).

Following recommendations by Rubin and Rubin [71], the combination of these approaches gave us the flexibility to organize categories into a hierarchy. For example, there were cases in which multiple subthemes fit into a larger overarching theme. Some participants' responses fit under multiple themes and were recorded accordingly. Responses from all identified subthemes did not necessarily add up to the total number of participants who fit into an overarching theme. Although some participants' answers fit into multiple themes, no two quotes used to illustrate a theme come from the same participant. When utilizing quotations, we corrected minor grammatical errors for the sake of clarity.

## Results

Analyses revealed that 52.6% of participants reported no change in religious devotion, 34.5% reported an increase in religious devotion, and 12.9% reported a decrease in religious devotion.

### Quantitative analyses

**Sociodemographics.**   We first examined the effect of group (decreased in religious devotion, increased in religious devotion, and no change) on sociodemographic variables in a MANOVA. There was no effect of condition (Wilk's Lambda = .98, $F$ (12.00, 1316.00) = 1.17, $p$ = .30) so we did not analyze any individual ANOVAs or post-hoc Bonferroni tests. As an additional preregistered robustness check, we initially planned to select control variables for supplemental analyses based upon any significant continuous covariates from the sociodemographic variables in the original MANOVA analyses. However, there were no significant sociodemographic differences in the original MANOVA analyses, so we did not carry out these supplemental analyses.

In addition to the MANOVA, gender, relationship status, and ethnicity were categorical sociodemographic variables that could not be utilized in a MANOVA analysis. Utilizing chi-square analyses, we tested whether there were overall differences between group. If any overall differences were identified, we then tested specific differences between groups. There was no significant difference for gender ($\chi^2$ (2) = .87, $p$ = .87). For relationship status, the only significant difference was between those who are single versus in a committed relationship ($\chi^2$ (2) = 7.24, $p$ = .03), with those who decreased being more likely to be in a committed relationship, as opposed to single, than those with no change ($\chi^2$ (1) = 4.65, $p$ = .03) and those who increased ($\chi^2$ (1) = 7.10, $p$ < .01). We also found a significant difference for ethnicity ($\chi^2$ (2) = 9.27, $p$ =

.01), with Black participants being more likely than White participants to increase in religious devotion than have no change ($\chi^2$ (1) = 9.28, $p < .01$).

**Religion variables.** We also conducted MANOVA analyses on religion variables. MANOVA analyses revealed a significant difference for the three groups' religious behaviors before COVID-19, Wilk's Lambda = .87, $F$ (8.00, 1324.00) = 12.35, $p < .001$. Those who reported an increase in religious devotion reported that they went to religious services more frequently, prayed more frequently, talked about religion, and read sacred texts more frequently than those who reported no change prior to COVID-19 (Table 1). In examining religious practices after the onset of COVID-19, Wilk's Lambda = .78, $F$ (8.00, 1338.00) = 22.04, $p < .001$, those who reported an increase in religious devotion engaged in all of these religious activities after COVD-19 more frequently compared to those who reported a decrease or no change. Finally, to examine if religious practices actually changed from pre-COVID to the onset of COVID, we created difference scores by subtracting religious practices before COVID from practices after COVID. Here too, we found significant differences for change in religious practices, Wilk's Lambda = .68, $F$ (8.00, 1310.00) = 35.26, $p < .001$. Reporting a decrease, staying the same, or increasing in religious devotion coincided with participants' report of prayer and reading spiritual texts (Table 1). All participants, however, reported a decrease in religious attendance, with the decrease group reporting the sharpest decrease in attendance. Additionally, there were significant differences for gratitude to God and religious coping, Wilk's Lambda = .85, $F$ (4.00, 1192.00) = 25.86, $p < .001$, with the increase group having higher gratitude to God and religious coping than the other two groups, and the decrease group having higher religious coping than the no change group.

There were, however, some variables that violated MANOVA normality of variance assumptions, shown through Levene's test, and required follow-up non-parametric Kruskal-Wallis H-tests: talking about religion and praying *before* COVID-19, attending religious service and praying *after* COVID-19, all four variables for changes in religious behavior, gratitude to God, and religious coping. We continued to find significant differences for all variables listed. With follow-up Mann-Whitney tests, results from the non-parametric tests were generally similar to the MANOVA results. The only exception is that, in the MANOVA based analysis, talking about religion before COVID-19 had a significant difference between the decrease and no change groups, but the difference was not found in the non-parametric based analysis. All results for non-parametric follow-ups are shown in Table 2.

## Individual differences

We also examined the effect of group (decreased in religious devotion, increased in religious devotion, and no change) on individual differences (e.g., Big 5 traits, comfort with uncertainty) in a MANOVAS. Similar to the sociodemographic variables, there was no effect of condition (Wilk's Lambda = .97, $F$ (14.00, 1192.00) = 1.39, $p = .15$) so we did not analyze any individual ANOVAs. Notably, it is less likely that effects for our other measures are due to a third variable such as differences in personality traits between groups given these null effects.

**Prosocial emotions.** There was also significant difference in reported prosocial emotions among the three groups (Wilk's Lambda = .96, $F$ (4.00, 1290.00) = 6.12, $p < .001$). Those who reported an increase in religious devotion reported higher general gratitude and awe than those who reported no change. They also reported higher gratitude than those who decreased in religious devotion (Table 1).

**Well-being.** Additionally, there was a significant difference in reported well-being among the three groups (Wilk's Lambda = .96, $F$ (6.00, 1280.00) = 4.77, $p < .001$). Participants who reported an increase or decrease in religious devotion reported higher search for meaning

**Table 1.** MANOVA results for changes in religious devotion.

| Variable | (a) Decrease | | (b) No Change | | (c) Increase | | Response Range | f Values | Partial Eta Squared | Wilk's Lambda |
|---|---|---|---|---|---|---|---|---|---|---|
| | M | SD | M | SD | M | SD | | | | |
| **Sociodemographic** | | | | | | | | | | .98 |
| Education | 4.84 | 1.32 | 4.45 | 1.46 | 4.52 | 1.35 | 1–7 | 2.65 | .01 | |
| Work hours | 25.81 | 18.80 | 24.84 | 20.53 | 25.20 | 19.50 | | .09 | .00 | |
| Number of kids | 1.02 | 1.05 | 1.05 | 1.39 | 1.14 | 1.19 | | .44 | .00 | |
| Social class | 2.84 | .77 | 2.79 | .83 | 2.77 | .85 | 1–5 | .21 | .00 | |
| Income level | 4.37 | 2.81 | 4.03 | 2.29 | 4.02 | 2.25 | 0–11 | .80 | .00 | |
| Political affiliation | 4.37 | 2.58 | 5.18 | 2.54 | 4.97 | 2.74 | 1–10 | 3.19 | .01 | |
| **Religion** | | | | | | | | | | |
| *Before COVID-19* | | | | | | | | | | |
| Religious service | 4.49[b] | 2.18 | 2.97[ac] | 2.07 | 4.13[b] | 2.16 | 0–7 | 30.01*** | .08 | .87*** |
| Talk | 4.32[b] | 2.44 | 3.65[ac] | 2.18 | 4.63[b] | 2.16 | 0–7 | 14.25*** | .04 | |
| Pray | 5.54 | 2.49 | 5.11[c] | 2.67 | 6.29[b] | 2.17 | 0–7 | 15.66*** | .05 | |
| Reading | 3.34[c] | 2.41 | 3.20[c] | 2.42 | 4.58[ab] | 2.49 | 0–7 | 23.15*** | .07 | |
| *After COVID-19* | | | | | | | | | | |
| Religious service | 2.31[c] | 1.93 | 2.25[c] | 1.92 | 3.61[ab] | 2.58 | 0–7 | 29.15*** | .08 | .78*** |
| Talk | 3.14[c] | 2.10 | 3.62[c] | 2.25 | 5.49[ab] | 2.16 | 0–7 | 62.55*** | .16 | |
| Pray | 4.48[c] | 2.63 | 5.12[c] | 2.66 | 6.98[ab] | 1.80 | 0–7 | 55.33*** | .14 | |
| Reading | 2.79[c] | 2.22 | 3.21[c] | 2.45 | 5.51[ab] | 2.50 | 0–7 | 73.96*** | .18 | |
| *Change in behavior* | | | | | | | | | | |
| Religious service | -2.29[bc] | 2.04 | -.72[a] | 1.42 | -.52[a] | 2.42 | -7-7 | 28.09*** | .08 | .68*** |
| Talk | -1.24[bc] | 1.84 | -.01[ac] | 1.04 | .88[ab] | 1.76 | -7-7 | 70.33*** | .18 | |
| Pray | -1.08[bc] | 1.71 | .03[ac] | .75 | .70[ab] | 1.27 | -7-7 | 82.65*** | .20 | |
| Reading spiritual text | -.63[bc] | 1.54 | .01[ac] | .61 | .96[ab] | 1.70 | -7-7 | 66.34*** | .17 | |
| *Other* | | | | | | | | | | |
| Gratitude to God | 7.05[c] | 1.67 | 7.08[c] | 1.65 | 8.04[ab] | 1.16 | 1–9 | 28.40*** | .09 | .85*** |
| Religious coping | 4.67[bc] | 1.61 | 4.03[ac] | 1.70 | 5.40[ab] | 1.43 | 1–7 | 46.41*** | .14 | |
| **Individual Differences** | | | | | | | | | | .97 |
| Need for closure | 4.60 | .89 | 4.46 | .92 | 4.58 | .85 | 1–7 | 1.38 | .01 | |
| Loneliness | 2.12 | .81 | 2.02 | .81 | 2.04 | .80 | 1–4 | .46 | .00 | |
| Extroversion | 3.95 | 1.59 | 3.75 | 1.54 | 3.97 | 1.60 | 1–7 | 1.41 | .01 | |
| Agreeableness | 5.36 | 1.07 | 5.46 | 1.10 | 5.63 | 1.09 | 1–7 | 2.21 | .01 | |
| Consciousness | 5.43 | 1.32 | 5.69 | 1.15 | 5.76 | 1.11 | 1–7 | 2.24 | .01 | |
| Neuroticism | 3.11 | 1.45 | 2.90 | 1.52 | 2.86 | 1.43 | 1–7 | .79 | .00 | |
| Openness | 5.30 | 1.08 | 5.04 | 1.25 | 5.20 | 1.24 | 1–7 | 1.96 | .01 | |
| **Prosocial Emotions** | | | | | | | | | | .96*** |
| Gratitude | 5.63[c] | .96 | 5.69[c] | 1.05 | 6.00[ab] | .92 | 1–7 | 7.86*** | .02 | |
| Awe | 4.70 | 1.14 | 4.63[c] | 1.26 | 5.07[b] | 1.21 | 1–7 | 8.99*** | .03 | |
| **Well-Being** | | | | | | | | | | .96*** |
| Life satisfaction | 4.26 | 1.32 | 4.39 | 1.50 | 4.50 | 1.38 | 1–7 | .89 | .00 | |
| Search for meaning | 4.97[b] | 1.41 | 4.42[ac] | 1.51 | 4.85[b] | 1.44 | 1–7 | 7.99*** | .02 | |
| Presence of meaning | 4.64[c] | 1.42 | 4.88 | 1.38 | 5.11[a] | 1.33 | 1–7 | 4.09* | .01 | |
| **COVID-19 Items** | | | | | | | | | | .79*** |
| *Unique Constructs* | | | | | | | | | | |
| Stress | 4.94[b] | 1.70 | 3.87[ac] | 1.78 | 4.48[b] | 1.71 | 1–7 | 13.89*** | .05 | |
| God protection | 4.06[c] | 2.30 | 3.71[c] | 2.20 | 5.41[ab] | 1.77 | 1–7 | 40.60*** | .13 | |

*(Continued)*

**Table 1.** (*Continued*)

| Variable | (a) Decrease M | SD | (b) No Change M | SD | (c) Increase M | SD | Response Range | f Values | Partial Eta Squared | Wilk's Lambda |
|---|---|---|---|---|---|---|---|---|---|---|
| Soc dis behaviors | 5.73 | .89 | 5.38[c] | 1.19 | 5.71[b] | 1.08 | 1–7 | 6.14** | .02 | |
| Prosocial behaviors | 3.08 | 1.43 | 2.81[c] | 1.31 | 3.46[b] | 1.44 | 1–7 | 13.84*** | .05 | |
| *COVID-19 Regulation Beliefs* | | | | | | | | | | |
| Soc dis morality | 5.69[b] | 1.57 | 4.97[ac] | 1.90 | 5.68[b] | 1.76 | 1–7 | 10.75*** | .04 | |
| Economy | 3.37 | 1.91 | 3.93 | 1.97 | 3.68 | 1.79 | 1–7 | 2.77 | .01 | |
| *Perceived Threat of COVID-19* | | | | | | | | | | |
| Health | 5.34[b] | 1.36 | 4.76[ac] | 1.62 | 5.38[b] | 1.53 | 1–7 | 9.70*** | .04 | |
| Personal finances | 5.18[b] | 1.61 | 4.26[ac] | 1.85 | 4.87[b] | 1.80 | 1–7 | 10.75*** | .04 | |
| Economy | 5.88 | 1.25 | 5.83 | 1.22 | 6.08 | 1.15 | 1–7 | 2.51 | .01 | |
| *Soc Dis Motives* | | | | | | | | | | |
| Avoid sickness | 6.09[b] | 1.08 | 5.65[ac] | 1.41 | 5.96[b] | 1.35 | 1–7 | 4.85** | .02 | |
| Avoid punishment | 4.00 | 1.77 | 3.79[c] | 1.77 | 4.22[b] | 1.75 | 1–7 | 3.53* | .01 | |

\* $p < .05$

\*\* $p < .01$

\*\*\* $p < .001$

Post-hoc tests utilized the Bonferroni method

Change in practice was calculated by subtracting religious practice before COVID-19 from religious practice after COVID-19

Superscripts indicate significant difference at the .05 level. For example, if the column "a" includes a superscript of "b" it indicates the value in column "a" is significantly different than the value in column "b".

Soc dis = Social Distancing

than those who reported no change. In addition, those who experienced an increase in religious devotion reported higher presence of meaning than those who reported a decrease. We did not find differences among the three groups in life satisfaction (Table 1).

**COVID-19 attitudes and behaviors.** Finally, we found a significant difference between the three groups on COVID-19 attitudes and behaviors (Wilk's Lambda = .79, *F* (22.00, 1092.00) = 6.33, *p* < .001). Therefore, we examined the underlying ANOVAS and if they were significant, applied a Bonferroni post-hoc test to assess pairwise comparisons. Participants who increased and decreased in religious devotion reported more stress and perceived a greater threat to their health from COVID-19 than those who had no change in religious devotion. Those who increased in their religious devotion had a stronger belief that God protects them from COVID-19 than did those in the decrease or no change groups. They also engaged in more social distancing behaviors and prosocial behaviors than those in the no change group. Both those who increased and decreased in religious devotion reported a higher belief that violating social distancing is morally wrong than those in the no change group, and also believed more strongly that COVID-19 was a threat to their health and personal finances than did those in the no change group. Finally, both the decrease in increase groups reported higher motivation to social distance to avoid sickness than the no change group, and the increase group reported higher motivation to social distance to avoid punishment than the no change group (Table 1).

For variables that violated Levene's test (i.e., God protects from COVID-10, social distancing behaviors, belief that violating social distancing policies is morally wrong) we followed up with non-parametric tests. Using the Kruskal Wallis test, believing God will protect them from COVID-19, *H* (2) = 82.63, *p* < .001, social distancing behaviors, *H* (2) = 8.25, *p* = .02, and believing it is morally wrong to violate social distancing policies, *H* (2) = 26.98, *p* < .001) were

**Table 2. Non-parametric test results for variables that violated normality assumptions.**

|  | (a) Decrease | | (b) No Change | | (c) Increase | |  |  |
|---|---|---|---|---|---|---|---|---|
| Variable | *M* | *SD* | *M* | *SD* | *M* | *SD* | Response Range | H value |
| **Religion** |  |  |  |  |  |  |  |  |
| *Before COVID-19* |  |  |  |  |  |  |  |  |
| Talk | 4.32 | 2.44 | 3.65[c] | 2.18 | 4.63[b] | 2.16 | 0–7 | 31.45*** |
| Pray | 5.54 | 2.49 | 5.11[c] | 2.67 | 6.29[b] | 2.17 | 0–7 | 27.87*** |
| *After COVID-19* |  |  |  |  |  |  |  |  |
| Religious service | 2.31[c] | 1.93 | 2.25[c] | 1.92 | 3.61[ab] | 2.58 | 0–7 | 46.67*** |
| Pray | 4.48[c] | 2.63 | 5.12[c] | 2.66 | 6.98[ab] | 1.80 | 0–7 | 9780*** |
| *Change in behavior* |  |  |  |  |  |  |  |  |
| Religious service | -2.29[bc] | 2.04 | -.72[a] | 1.42 | -.52[a] | 2.42 | -7-7 | 52.09*** |
| Talk | -1.24[bc] | 1.84 | -.01[ac] | 1.04 | .88[ab] | 1.76 | -7-7 | 121.91*** |
| Pray | -1.08[bc] | 1.71 | .03[ac] | .75 | .70[ab] | 1.27 | -7-7 | 130.39*** |
| Reading spiritual text | -.63[bc] | 1.54 | .01[ac] | .61 | .96[ab] | 1.70 | -7-7 | 108.65*** |
| Gratitude to God | 7.05[c] | 1.67 | 7.08[c] | 1.65 | 8.04[ab] | 1.16 | 1–9 | 61.48*** |
| Religious coping | 4.67[bc] | 1.61 | 4.03[ac] | 1.70 | 5.40[ab] | 1.43 | 1–7 | 90.47*** |
| **COVID-19 Items** |  |  |  |  |  |  |  |  |
| God protects from | 4.01[c] | 2.31 | 3.69[c] | 2.19 | 5.41[ab] | 1.76 | 1–7 | 82.63*** |
| COVID |  |  |  |  |  |  |  |  |
| Social distancing | 5.71 | .89 | 5.38[c] | 1.19 | 5.71[b] | 1.08 | 1–7 | 8.25* |
| behaviors |  |  |  |  |  |  |  |  |
| *Covid-19 Regulation* |  |  |  |  |  |  |  |  |
| *Beliefs* |  |  |  |  |  |  |  |  |
| Soc dis morality | 5.71[b] | 1.56 | 4.97[ac] | 1.90 | 5.68[b] | 1.76 | 1–7 | 26.98*** |

*$p < .05$

** $p < .01$

*** $p < .001$

Whitney-Mann Tests were used as post-hoc tests to evaluate direct comparisons between groups

Change in practice was calculated by subtracting religious practice before COVID-19 from religious practice after COVID-19

Superscripts indicate significant difference at the .05 level. For example, if the column "a" includes a superscript of "b" it indicates the value in column "a" is significantly different than the value in column "b".

Soc dis = Social Distancing

all significant. With follow-up Mann-Whitney tests, results from the non-parametric tests were similar to those found when using the post-hoc Bonferroni tests from the MANOVA based analysis.

*Summary*. In summary, our results revealed that changes in prayer and reading sacred texts corresponded with changes in religious devotion, but religious service attendance decreased for all groups, especially those who decreased in religious devotion. Those who increased and decreased in religious devotion were more likely than those with no change to report greater stress and threat from COVID-19, but only those who increased in religious devotion reported more prosocial emotions like gratitude and awe. In addition, those who increased and decreased in religious devotion were more likely to be searching for meaning than those with no change, but only those who increased in religious devotion showed higher presence of meaning, in comparison to those in the other groups. These findings suggest that more negative and tense states like stress, threat, and searching for meaning characterize those who changed in religious devotion, but for those who increased, these states were also paired with more positive experiences like greater prosocial emotions and greater presence of meaning.

## Qualitative analyses

Having just documented the broad differences between those who increased, stayed the same, or decreased in religious devotion, we now provide deeper insight into the reasons for these changes with participants' open ended responses accounting for what they believed were the reasons for their change or lack of change in religious devotion.

**Increase group.** We coded the responses of participants who increased in religious devotion ($N$ = 237) into three overarching themes and seven subthemes (Table 3).

*Engagement in personal worship.* A common theme we noted among those who reported an increase in religious devotion was personal worship (n = 113). Participants indicated

**Table 3. Qualitative findings on why participants experienced changes in religious devotion.**

| | Main Themes | Subthemes |
|---|---|---|
| Increase (n = 237) | Engagement in Personal Worship (n = 113) "I have found much more comfort with the current situations by praying and being closer to God" | Prayer (n = 82) "I find myself praying more" |
| | | Scripture Study (n = 23) "Continue to study the Word daily" |
| | Need for Higher Power (n = 100) "I tend to rely on God in the tough moments when I don't understand why something bad is happening" | Alleviate Distress and Find Peace in Trying Times (n = 42) "I'm scared and prayer makes me feel calm" |
| | | Faith in God Will Overcome the Virus (n = 31) "I'm praying more that this virus goes away quickly" |
| | | Sense of Control and Security (n = 15) "I find security in higher power" |
| | Uncertainty in Life (n = 54) "Learning more overtime how important it is to spend time with God" | Death is Close (n = 15) "I'm afraid to die" |
| | | Gratitude and Appreciation (n = 12) "I appreciate the things more that God has given to me" |
| No Change (n = 354) | Same Religious Faith (n = 174) "I am still practicing as I normally do" | |
| | Not Religious (n = 119) "I didn't practice before, and it hasn't really changed" | |
| | Vague Responses (n = 46) "Have other things to do with my time, have other interests" | |
| Decrease (n = 90) | Lack of Engagement in Communal Worship (n = 62) "The inability to go to church and worship with my brethren is very taunting" | Not Being Able to Attend Religious Services (n = 55) "Local churches and services have been canceled" |
| | | Mode of Delivery (n = 23) "It's harder to stay connected when everything is online" |
| | Lack of Commitment or Priority (n = 14) "Not really attending or focusing on that right now. Work and family are key" | No Time for Religiosity (n = 8) "Less time in church, more time to think about other things" |
| | | Experience of Boredom and Laziness (n = 6) "Not going to church makes me lazy to study the Bible and pray" |
| | Hardship Challenging Belief in God (n = 9) "It is hard to believe that if God exists, he would let this happen" | |

strengthening their religious devotion through their worship to God through different religious activities. Some participants made statements that broadly fit this theme, including "learning more over time how important it is to spend time with God." In increasing their personal worship, the most frequent religious behavior participants reported increasing was prayer (n = 82), writing that they "pray more multiple times a day for myself and others to get through this pandemic." Participants also reported increasing was scripture study (n = 23) in which they reported things such as "more time to do my quiet time, read more and reflect on my Bible."

*Need for a higher power.* Another reason why participants reported an increase was feeling the need for a higher power to support them and help them go through the hardships and stress of COVID-19 (n = 100). Participants said things that highlighted how God helped them cope with the perceived stress of COVID-19 such as the "pandemic has changed our lives in so many ways. I feel like I need God more than ever before to get through these difficult times." More specifically, participants reported that turning to this higher power helped them alleviate distress and find peace in trying times (n = 42), stating that "My strong religious beliefs have given me comfort and peace during this difficult time." Some participants believed that their faith in God will help overcome the virus (n = 31), believing that God "has power to save us from this pandemic." Finally, some participants reported that their increased religious devotion gave them a sense of control and security over the situation (n = 15), saying things like, "I think it made me realize that it is one of the only secure and consistent things in my life."

*Uncertainty in life.* Participants also reported increasing in religious devotion because they felt "tomorrow is not promised," and idea that the future is uncertain, and death may be near (n = 54). Some participants reported that "It [COVID-19] has made me more aware of the most important things in life." Overall participants felt the urge to "reassess priorities." They realized that death was close after seeing thousands of people dying around the world (n = 15). They mentioned having "more time to myself overall to think about God and my relationship with him and what will happen when I die." Others experienced gratitude and appreciation for days when they and their families wake up healthy, but also for the small things in their lives (n = 12), such as the people surrounding them: "Reading about what other people have gone through due to COVID-19, I am grateful for my life and people in it."

**Decrease group.** A second group of people who reported a decrease in religious devotion (n = 90) were coded into three overarching themes and four subthemes (Table 3).

*Lack of engagement in communal worship.* One of the main reasons reported by participants for their decrease in religious devotion was the lack of worship (n = 62), meaning that they were not on track with their usual religious activities, thus, leading to a natural decline in devotion. This lack of worship was manifested mainly by not being able to attend religious services (n = 55). Participants felt that religion was not the same without the sense of community. One reason for this is the change in mode of delivery (n = 23), with religious services being shifted to online platforms, participants were "not as inclined to 'attend' mass if I'm just watching it, on TV." In contrast to those in the increase group, three participants reported that they had declined in religious devotion because of a lack of personal worship such as they "have not taken enough time in study and prayer."

*Lack of commitment or priority.* Another reason for reporting a decline in devotion was a lack of commitment (n = 14). Some participants reported having no time for religiosity (n = 8), saying things like, "Not really attending or focusing on that right now. Work and family are key." Another reported, "I have not been following up with religious activities at all. I think that I'm more concerned about the pandemic." Another reason was experiencing boredom and laziness (n = 6), feeling, "I don't have the motivation to attend virtual services."

*Hardship challenging belief in God.* Perceived hardship during COVID-19 was another reason participants decreased in their religious devotion (n = 9) since this hardship challenged

their belief in God. Participants mentioned that it was "hard to maintain a religious belief system when there is so much suffering" and "it is hard to believe that if God exists, he will let this happen."

**No change group.**   A third group of people reported no change in religious devotion (n = 354). First, some people reported being religious before COVID-19 and were still experiencing the same level of devotion (n = 174). Second, people reported not being particularly religious before COVID-19, which stayed the same after COVID-19 (n = 119). A final group responded unclearly or vaguely n = 46). Examples are found in Table 3.

## Discussion

The primary aim of our study was to understand the characteristics of those who reported change versus stability in religious devotion as a result of the COVID-19 pandemic. We assessed how changes in religious devotion are reflected in sociodemographics, religion variables, individual differences, prosocial emotions, well-being, and COVID-19 specific attitudes and behaviors. We also gave participants the opportunity to explain, in their own words, *why* they increased, stayed the same, or decreased in religious devotion since the onset of COVID-19. Despite some research suggesting potential differences based on sociodemographics and individual differences, we did not find any differences in our dataset. Potential reasons are highlighted in our limitations section. In synthesizing quantitative and qualitative results, we present results in the order of COVID-19 attitudes and behaviors, religion variables, prosocial emotions, and well-being.

For COVID-19 attitudes and behaviors, those who increased in religious devotion had some unique outcomes such as being most likely to believe God would protect them from COVID-19 and engaging in more COVID-19 prosocial behaviors. But perhaps the biggest overall takeaway from these many variables is that both those who increased and decreased in religious devotion were generally more likely to experience greater stress and threat from COVID-19. The potential destabilization of a worldview that can come from perceiving events to be highly stressful [32] may be part of the reason why both those who increased and decreased in religious devotion reported higher searching for meaning. Overall, the commonalities between those who decreased and increased in religious devotion support previous research suggesting that experiencing greater stress during a crisis may lead to a higher likelihood of trying to make sense of the event by searching for meaning [53] and turning toward [32] or away [6] from God. Our qualitative findings similarly suggest that people who increased in religious devotion did so to alleviate stress and those who decreased in religious devotion cited reduced time to engage in religious activities due to managing life in a pandemic.

Efforts to alleviate their perceived threat of COVID-19 were also based in how individuals engaged in religious practices. Both quantitative and qualitative results shed light on what religious behaviors were actually changing when participants reported an increase or decrease in religious devotion. Those who decreased in their religious devotion reported less engagement in all three religious behaviors (i.e., attending religious services, praying, reading sacred texts) after COVID-19 in comparison to their retrospective report of before COVID-19. Unsurprisingly, likely due to restrictions from attending worship services and social distancing policies, attending religious services had the sharpest drop in comparison to the other behaviors. Even those who reported an increase in religious devotion reported lower attendance of worship services after COVID-19 than before, despite increasing in the other religious behaviors. Our qualitative data, in particular, seem to speak to this distinction between focusing on more internal versus external religiosity. Among those who reported an increase in religious devotion, 47.7% participants focused their comments specifically on engaging in personal worship

such as prayer or scripture study; alternatively, among those who reported a decrease in religious devotion, 68.9% mentioned the lack of engagement in communal worship. This difference is striking given that we did not specifically prompt participants to report on these experiences denoting internal or external religiosity. Rather they spontaneously reported them in response to a broad, open-ended question about "Why" their religious devotion had changed or not changed since COVID-19. This is broadly consistent with coping theory suggesting that ritualistic religious behaviors can help alleviate distress by providing a sense of meaning amidst a crisis [32], which also seemed reflected in quantitative results showing that those who increased in religious devotion reported higher positive religious coping and gratitude to God. This is also consistent with previous research specifically suggesting increases in intrinsic religiosity following a crisis [36]. The appeal of more private internal religiosity may be particularly strong due to social distancing regulations. It appears that those who decreased in religious devotion were more community oriented in their religious devotion and did not have a community to turn to after COVID-19 due to regulations surrounding worship services.

Despite greater reported stress and threat from COVID for those who changed in religious devotion, only those who increased were higher in the prosocial emotions of gratitude and awe. Greater gratitude also appeared with the theme of "Gratitude and Appreciation" for those who increased in religious devotion. There could be something unique about the religious behaviors in which people engage that reinforce dispositions that help provide meaning. For example, both quantitative and qualitative data showed that prayer was a specific religious behavior that those who increased in religious devotion engaged in most frequently. Prayer can induce self-transcendent experiences that, in some cases, directionally predict and in other cases experimentally lead to gratitude [47] and awe [50, 72], This is consistent with broader coping theory suggesting that religion can help people regulate their emotions, by both providing ritualistic actions that can alleviate negative emotions and promote prosocial emotions [32].

Consistent with previous research [52], both our quantitative and qualitative results suggest that while those who increased and decreased in religious devotion were searching for meaning, only those who increased in religious devotion reported finding the meaning they were searching for. Some of the results found in the qualitative data indirectly converge on this point as well. For those who decreased in religious devotion, one theme that emerged was that hardship was challenging their belief in God. As religion and understanding of God can be a source of meaning in life, it follows that the shaking of this meaning structure would result in people struggling to find meaning in the face of a crisis. The qualitative data also had several themes that gave voice to those who increased in religious devotion and were successfully finding meaning. The theme "Uncertainty in Life" showed that those who increased in religious devotion were reflecting upon bigger life questions and trying to make sense of what was happening. Rather than reporting a challenge to their belief in God, they reported other elements that seemed to suggest developing a meaningful coherence from COVID-19 as a stressor. They reported that their need for a higher power helped them "alleviate distress and find peace", that "faith in God will overcome the virus" and that their faith provided a "sense of control and security." Consistent with coping theory, feelings of peace, belief in a brighter future, and gaining a sense of control can help to stabilize an individual in a stressful situation, help them interpret life events, and maintain a sense of meaning in life [32].

## Limitations and future directions

Although our study had strengths such as a multi-method approach and a nationally representative sample of participants recruited shortly after the onset of COVID-19 related regulations,

it also had limitations. One limitation is attempting to understand change with a cross-sectional design. Although we have noted a plausible direction for how these variables connect with each other, future research could more closely examine the extent to which predispositions lead participants to interpret events in a way that increases religious devotion, the extent to which specific religious behaviors may cultivate dispositions that help people to cope with stressful life events, as well as the extent to which there is some sort of bidirectionality between the two. The cross-sectional nature of the data may be part of the reason we did not detect several significant group differences for changes in religious devotion for hypothesized connections from previous research. For example, previous research has shown that religion can buffer against the detrimental effects of neuroticism [73]. However, valuable information on this connection may be washed out in the crosscurrents of cross-sectional research. For example, some participants who score higher in neuroticism may report an increase in religious devotion because they were seeking an avenue to alleviate their anxiety. However, some who report an increase in religious devotion might report lower neuroticism because they feel religion has alleviated their anxiety. If both of these effects are occurring in tandem, they could result in no significant differences between those who increased in religious devotion and those who decreased or exhibited no change. Similar concerns may come into play for other sociodemographic and individual difference variables, as there were no significant differences between groups for any of these variables. It would be beneficial to obtain longitudinal data to assess how conditions (e.g., personality traits) before a stressor might predict outcomes after the stressor.

Additionally, while there are advantages to understanding the unique circumstances surrounding religious change during a universal stressor such as COVID-19, there are some factors that make it challenging to know how the findings might be integrated into the wider literature on religion and coping [32]. For example, the uniqueness of social distancing may have resulted in some findings unique to this particular stressor and questionable how well they apply to the wider literature on stress and religion as a coping mechanism. Of note, the finding that even those who increased their religious devotion did not increase their worship service attendance could be unique to COVID-19 regulations.

Another note is that our data were collected in May 2020, in the early stages of the pandemic, which has the distinct advantage of allowing us to capture immediate effects of COVID-19 on religious devotion. However, the influence of the pandemic is long and pervasive, and we cannot be sure of the extent to which changes will endure over time. Previous research on the 9/11 attacks suggests that some of the immediate effects of a major stressor might become weaker over time [74, 75]. Based on this work, it would be beneficial for researchers to assess the extent some of these religious changes are immediate as opposed to longer-term. It would also be beneficial to assess specific differences between different religious affiliations, as the majority of our sample was Christian.

Finally, we note that although there were not significant differences between groups for sociodemographic variables, that the means for those in the decrease group seemed to suggest higher education, fewer kids, higher income, belonging to a higher social class, and being liberal. As these variables are connected to secularization of Western countries, it could be valuable for future research to more closely explore whether those who are already susceptible to a decline in the intensity of their faith may have done so due to COVID-19 related stress and demands.

## Conclusion

The suffering, loneliness, and stagnation brought on by the COVID-19 pandemic has challenged many to re-evaluate existential questions about life's meaning and one's purpose.

Religion is one suitable option to help individuals navigate this complex and stressful experience. Our work suggests that those who turn to God find a greater sense of meaning and experience emotions like gratitude and awe that bind them to others, which may help combat the stress and threat of the pandemic. While some of the communal aspects of religion have suffered, personal worship may support people's religious connection through this difficult time.

## Supporting information

**S1 Table. Exploratory factor analysis for COVID-19 belief items.** Factors are indicated by bold values.
(DOCX)

**S2 Table. Exploratory factor analysis for COVID-19 social distance motivation items.** Factors are indicated by bold values.
(DOCX)

**S1 Fig. Eigenvalue scree plot for COVID-19 belief items.**
(DOCX)

**S2 Fig. Eigenvalue scree plot for COVID-19 social distancing motivation items.**
(DOCX)

## Author Contributions

**Conceptualization:** Nathan D. Leonhardt, Sarah Fahmi, Jennifer E. Stellar, Emily A. Impett.

**Data curation:** Nathan D. Leonhardt, Sarah Fahmi.

**Formal analysis:** Nathan D. Leonhardt, Sarah Fahmi.

**Funding acquisition:** Emily A. Impett.

**Investigation:** Nathan D. Leonhardt.

**Methodology:** Nathan D. Leonhardt, Sarah Fahmi, Jennifer E. Stellar, Emily A. Impett.

**Supervision:** Jennifer E. Stellar.

**Writing – original draft:** Nathan D. Leonhardt, Sarah Fahmi.

**Writing – review & editing:** Nathan D. Leonhardt, Sarah Fahmi, Jennifer E. Stellar, Emily A. Impett.

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
