## [Decision Letter · Decision Letter 0]

14 Mar 2022

PONE-D-22-03669Turning Toward or Away from God: COVID-19 and Changes in Religious DevotionPLOS ONE

Dear Dr. Leonhardt,

Thank you for submitting your manuscript to PLOS ONE. After careful consideration, we feel that it has merit but does not fully meet PLOS ONE’s publication criteria as it currently stands. Therefore, we invite you to submit a revised version of the manuscript that addresses the points raised during the review process.

We look forward to receiving your revised manuscript.

Kind regards,

Andrea Fronzetti Colladon, Ph.D.

Academic Editor

PLOS ONE

Journal Requirements:

Additional Editor Comments:

We have received the reports from our advisors on your manuscript and read them carefully. We think the reviewers provided very good assessments and recommendations. Regarding the content of the reviews, we find no inconsistencies and we share their comments. Based on the advice received, we feel that your manuscript could be reconsidered for publication should you be prepared to incorporate revisions.

You should appreciate that this is a *major* revision. Minor changes to your manuscript will not be acceptable. Furthermore, current decision does not guarantee an eventual acceptance of your paper - thus, the importance of your revisions.

Reviewers' comments:

Reviewer's Responses to Questions

**Comments to the Author**

1. Is the manuscript technically sound, and do the data support the conclusions?

Reviewer #1: Yes

Reviewer #2: Partly

2. Has the statistical analysis been performed appropriately and rigorously? 

Reviewer #1: Yes

Reviewer #2: No

3. Have the authors made all data underlying the findings in their manuscript fully available?

Reviewer #1: Yes

Reviewer #2: Yes

4. Is the manuscript presented in an intelligible fashion and written in standard English?

Reviewer #1: Yes

Reviewer #2: Yes

5. Review Comments to the Author

Reviewer #1: The authors conducted an exploratory study of how the COVID pandemic may have influenced religious change. They provide a good summary of their findings in the abstract and on page 15 of the manuscript. Of interest, people who were more intrinsically-oriented increased in prayer, reading, and devotion; whereas people who were more church-oriented (extrinsically-oriented) decreased in devotion (presumably, because many in-person church services were cancelled). People who did not change reported feeling less threat, less stress, and less religiousness – yet high life satisfaction and meaning in life.

The conclusions are mostly intuitive, adding only incrementally to our understanding of the effects of the pandemic on religiosity in the U.S. This might be improved by including a final path analysis with Threat from COVID being the exogenous (first) variable predicting Stress; Stress predicting (1) change in religious service attendance, (2) change in prayer, (3) change in reading religious literature; each type of change predicting each of the outcome variables, (1) life satisfaction, (2) meaning in life, (3) search for meaning, (4) gratitude, and (5) awe. A path model like this might allow us to further explore whether people are more likely to turn to religious attendance, prayer, or reading when faced with stress. We would also have a better idea of which type of these three religious practices was more likely to foster each of the positive outcomes (e.g., meaning in life) while controlling for all other correlations in the model.

I note here that the “threat to stress to religious change” path summarizes the argument in the introduction. However, the authors include stress as an outcome of religious change in the analyses and results. This highlights one of the many problems with assessing religious change in cross-sectional data (e.g., we can’t know if the people who did not change in devotion had less stress or whether their devotion reduced their stress level). To their credit, the authors do acknowledge this in the Discussion. As another example, with cross-sectional data, we must rely on introspection and self-report as to whether or not devotion actually changed. And people may not be very good at assessing (or reporting) their own level of devotion from Time 1 to Time 2.

Regarding preregistration, I would not agree that this study was preregistered. Although the authors refer the reader to a document listed as “preregistration” on OSF (osf.io/a4jcf/?view_only=a9e3e2b5d79d4ab3b9905e7106784b72), the document was prepared after the data were collected and after the qualitative data were analyzed; the percentages in each of the three groups has changed between the preregistration and manuscript; and there are some variables (e.g., religious coping and prosocial behavior) that are pertinent to the argument in the introduction of the manuscript that are not included in the analyses in the manuscript. Moreover, the pre-registration lists 11-items used to assess COVID stress and perceived threat. However, only five of the items were used for the analyses presented in the manuscript and those five items were separated into two variables. Overall, the “preregistration” and the manuscript analyses are so different that I must conclude the authors are overstating by claiming that this study (at least the quantitative analyses) were pre-registered. This does not preclude publication . . . I think there are real benefits to conducting exploratory research and exploratory research is suitable for publication. But the quantitative analyses were not preregistered, strictly speaking.

Finally, I want to add that the manuscript is very well written. However, references #2 through #6 (Geertz, Fromm, Freud, Buber, Durkheim) seem oddly out of place, inappropriate, and irrelevant in this paper. There are plenty of psychology papers that could have been cited to support the argument that religion helps people cope with life’s challenges. (And there are plenty of other papers not cited showing that religion can actually lead to negative coping and spiritual struggles!) Additionally, in terms of references cited, there are only three articles cited focusing specifically on the links between COVID and religion. However, there have been many articles published either supporting, elaborating, or contradicting the findings of this study that are not referenced. I have listed just a few of the 192 articles listed on PsycInfo at the end of this review. Although it is not necessary to cite all of the ones listed below, the authors should be familiar with and at least cite the most relevant.

I hope my comments can be helpful to the editor and authors.

Additional articles of interest:

Davis, E. B., McElroy-Heltzel, S., Lemke, A. W., Cowden, R. G., VanderWeele, T. J., Worthington, E. L., Jr., . . . Aten, J. D. (2021). Psychological and spiritual outcomes during the COVID-19 pandemic: A prospective longitudinal study of adults with chronic disease. Health Psychology, 40(6), 347-356.

DeRossett, T., LaVoie, D. J., & Brooks, D. (2021). Religious coping amidst a pandemic: Impact on COVID-19-related anxiety. Journal of Religion and Health, 60(5), 3161-3176.

Johnson, K. A., Baraldi, A. N., Moon, J. W., Okun, M. A., & Cohen, A. B. (2021). Faith and science mindsets as predictors of COVID-19 concern: A three-wave longitudinal study. Journal of Experimental Social Psychology, 96, 14.

Kasapoğlu, F. (2022). The relationship among spirituality, self-efficacy, covid-19 anxiety, and hopelessness during the covid-19 process in turkey: A path analysis. Journal of Religion and Health, doi:http://dx.doi.org/10.1007/s10943-021-01472-7

Kranz, D., Niepel, C., Botes, E., & Greiff, S. (2020). Religiosity predicts unreasonable coping with COVID-19. Psychology of Religion and Spirituality, doi:http://dx.doi.org/10.1037/rel0000395

Michaels, J. L., Hao, F., Ritenour, N., & Aguilar, N. (2022). Belongingness is a mediating factor between religious service attendance and reduced psychological distress during the covid-19 pandemic. Journal of Religion and Health, doi:http://dx.doi.org/10.1007/s10943-021-01482-5

Papazoglou, A. S., Moysidis, D. V., Tsagkaris, C., Dorosh, M., Karagiannidis, E., & Mazin, R. (2021). Spiritual health and the COVID-19 pandemic: Impacts on orthodox christianity devotion practices, rituals, and religious pilgrimages. Journal of Religion and Health, 60(5), 3217-3229.

Pirutinsky, S., Cherniak, A. D., & Rosmarin, D. H. (2020). COVID-19, mental health, and religious coping among american orthodox jews. Journal of Religion and Health, 59(5), 2288-2301.

Rigoli, F. (2021). The link between covid-19, anxiety, and religious beliefs in the united states and the united kingdom. Journal of Religion and Health, doi:http://dx.doi.org/10.1007/s10943-021-01296-5

Zacher, H., & Rudolph, C. W. (2021). Individual differences and changes in subjective wellbeing during the early stages of the COVID-19 pandemic. American Psychologist, 76(1), 50-62.

Zhang, H., Hook, J. N., Hodge, A. S., Coomes, S. P., Davis, C. W., Van Tongeren, D. R., . . . Aten, J. D. (2021). Religious and spiritual struggles and coping amidst the COVID-19 pandemic: A qualitative study. Spirituality in Clinical Practice, 8(4), 245-261.

Reviewer #2: The manuscript describes a correlational study conducted at the beginning of the Covid-19 pandemic on the effects of the pandemic on change in religious devotion. The results showed rather a trivial finding that some participants increased, some decreased in their religious devotion while others remained unchanged. A more nuanced look into the change in religiosity revealed that while an increase in personal prayer and engagement with the scripture was characteristic for the group with increased religiosity, the opposite was true for the group with decreased religiosity. A qualitative analysis further showed that increased religiosity was associated with increased stress and search for meaning in life while decreased religiosity with less time that could be possibly dedicated to religious activities.

Overall, this is a straightforward study with several strengths – pre-registered hypotheses, data coming from a crucial period during the pandemic, and a combination of qualitative and quantitative approaches. As I noted, the findings from this study are not overwhelming, but I find that there is merit to them, especially if the authors would be willing to put more work into the manuscript.

Below are several suggestions how the manuscript could be improved:

1) I found the intro a bit lengthy and repetitive. I believe that the writing could be more concise; there is a lot of repetition in the paper.

2) Notably, shortening the current manuscript will also allow including information that was moved to the supplementary material but should be included in the main text—namely, the theoretical rationale for all the pre-registered hypotheses and their results. When I was reading the introduction, I was surprised by the inclusion of the gratitude and awe measures that did not seem to follow from any theory. I immediately started to wonder how many measures were obtained and whether including these two measures in the introduction is just because these measures came out as statistically significant in the results section. Indeed, this is what the authors confirmed in the first paragraph on p. 15 where they noted the use of other measures that were moved to supplements because their results were not statistically significant. While I understand the reasons for this decision, I believe it harms the paper because the results feel cherry-picked, and the underlying theory is insufficient. Indeed, after reading the paper, I kept wondering why the authors did not try to address the most interesting question, that is, what predicts an increase or decrease in religiosity due to the Covid-19 pandemic? Looking into the pre-registration file revealed that the authors included several variables to this end. Thus, the introduction should state the theoretical rationale for including these variables and hypotheses, and the results should be reported in the main text. Likewise, the authors should discuss what does the lack of statistically significant results mean for the theory.

3) Looking further into the dataset the authors provided (thank you for sharing the data), I see that they collected many variables that could be used in an explorative analysis. For instance, it could be assessed whether the number of kids contributed to the decrease in religiosity (as people have less time to participate in religious activities). I will leave this up to the authors, but it seems like the data actually have more potential than discussed in the manuscript. I also feel like the sheer extent of the dataset deserves a comment in the main text.

4) Would it be possible to include also the original questionnaire in the OSF repository? I aimed to check some of the analyses conducted in the paper yet could not make sense of the dataset.

5) The reason I wanted to conduct more analyses myself was that I am worried about the correlational nature of the study and whether all major confounds were appropriately taken into account. Generally, MANOVA does not strike as the best analytical approach in this case since there was no random assignment to the experimental groups, and one cannot assume that the groups would be more or less homogenous due to the random assignment. The authors deal with this issue on p. 14 by conducting MANOVAs where increase/decrease of religiosity predicts sociodemographic variables, but this is not sufficient. A regression approach would be more appropriate in this situation since all the effects of covariates could be used to adjust the main investigated effects of religious increase/decrease on the outcome variables of interest. Likewise, the authors do not report any fit indices of their MANOVA models, but I wonder how the normality assumption held with dependent variables being often measured at Likert scales.

6) I believe that the work of Bronislaw Malinowski and his contemporary followers should be discussed in the manuscript since it could help explain some of the findings regarding the effects of stress and uncertainty. For example, see the two references below:

Lang, M., Krátký, J., & Xygalatas, D. (2020). The role of ritual behavior in anxiety reduction: An investigation of Marathi religious practices in Mauritius. Philosophical Transactions of the Royal Society B, 375(1805), 20190431. https://doi.org/https://doi.org/10.1098/rstb.2019.0431

Sosis, R., & Handwerker, W. P. (2011). Psalms and coping with uncertainty: Religious Israeli women’s responses to the 2006 Lebanon war. American Anthropologist, 113(1), 40–55. https://doi.org/10.1111/j.1548-1433.2010.01305.x

7) The fact that pre-registration was conducted after data collection should be noted in the paper (apologies if I have missed it)

Overall, I believe that after working on the manuscript a bit more, it would be a valuable contribution to the current literature on the relationship between religion and stressful life events.

6. PLOS authors have the option to publish the peer review history of their article (what does this mean?). If published, this will include your full peer review and any attached files.

Reviewer #1: No

Reviewer #2: No

---

## [Author Response · Author response to Decision Letter 0]

14 Sep 2022

Andrea Fronzetti Colladon, Ph.D.

Academic Editor

PLOS ONE

Dr. Colladon,

Thank you for your patience in giving us extra time to resubmit this manuscript. We have carefully incorporated feedback from you as an editor and feedback from both of the reviewers. Below we responded to each point of feedback.

Editor:

Comment: Please ensure that your manuscript meets PLOS ONE's style requirements, including those for file naming. The PLOS ONE style templates can be found at 

Response: We have reviewed the style requirements and have incorporated them into our submission. If there are still issues we have not addressed, we will be happy to correct them.

Comment: We note that the grant information you provided in the ‘Funding Information’ and ‘Financial Disclosure’ sections do not match. When you resubmit, please ensure that you provide the correct grant numbers for the awards you received for your study in the ‘Funding Information’ section.

Response: We reviewed the grant information and completed information for Funding Information in the submission portal. We are struggling to locate the Financial Disclosure section, but we will be happy to make sure that they match.

Comment: In your Data Availability statement, you have not specified where the minimal data set underlying the results described in your manuscript can be found. PLOS defines a study's minimal data set as the underlying data used to reach the conclusions drawn in the manuscript and any additional data required to replicate the reported study findings in their entirety. All PLOS journals require that the minimal data set be made fully available. For more information about our data policy, please see http://journals.plos.org/plosone/s/data-availability. Upon re-submitting your revised manuscript, please upload your study’s minimal underlying data set as either Supporting Information files or to a stable, public repository and include the relevant URLs, DOIs, or accession numbers within your revised cover letter. For a list of acceptable repositories, please see http://journals.plos.org/plosone/s/data-availability#loc-recommended-repositories. Any potentially identifying patient information must be fully anonymized. Important: If there are ethical or legal restrictions to sharing your data publicly, please explain these restrictions in detail. Please see our guidelines for more information on what we consider unacceptable restrictions to publicly sharing data: http://journals.plos.org/plosone/s/data-availability#loc-unacceptable-data-access-restrictions. Note that it is not acceptable for the authors to be the sole named individuals responsible for ensuring data access. We will update your Data Availability statement to reflect the information you provide in your cover letter.

Response: Our data are uploaded to a public repository on OSF. We have clarified this information in the revised manuscript.

Comment: Please include captions for your Supporting Information files at the end of your manuscript, and update any in-text citations to match accordingly. Please see our Supporting Information guidelines for more information: http://journals.plos.org/plosone/s/supporting-information. 

Response: Due to the requests of the reviewers, the Supporting Information files have been largely integrated into the main manuscript. The only remaining Supporting Information files are concerning the psychometrics of the COVID-19 focused items. We have ensured that those files are consistent with the journal recommendations.

Comment: We have received the reports from our advisors on your manuscript and read them carefully. We think the reviewers provided very good assessments and recommendations. Regarding the content of the reviews, we find no inconsistencies and we share their comments. Based on the advice received, we feel that your manuscript could be reconsidered for publication should you be prepared to incorporate revisions. You should appreciate that this is a *major* revision. Minor changes to your manuscript will not be acceptable. Furthermore, current decision does not guarantee an eventual acceptance of your paper - thus, the importance of your revisions.

Response: We appreciate the opportunity to revise the manuscript. We have carefully considered the feedback of the reviewers and have been able to incorporate the vast majority of suggestions. We have highlighted a few points that may be in need of additional clarity and hope for guidance on whether changes adequately address concerns or could use further refining.

 

Reviewer #1 

Comment: The authors conducted an exploratory study of how the COVID pandemic may have influenced religious change. They provide a good summary of their findings in the abstract and on page 15 of the manuscript. Of interest, people who were more intrinsically-oriented increased in prayer, reading, and devotion; whereas people who were more church-oriented (extrinsically-oriented) decreased in devotion (presumably, because many in-person church services were cancelled). People who did not change reported feeling less threat, less stress, and less religiousness – yet high life satisfaction and meaning in life.

Response: We appreciate the acknowledgement that the abstract provided a good summary of our findings. We hope that remains the case with the many changes we made to the manuscript.

Comment: The conclusions are mostly intuitive, adding only incrementally to our understanding of the effects of the pandemic on religiosity in the U.S. This might be improved by including a final path analysis with Threat from COVID being the exogenous (first) variable predicting Stress; Stress predicting (1) change in religious service attendance, (2) change in prayer, (3) change in reading religious literature; each type of change predicting each of the outcome variables, (1) life satisfaction, (2) meaning in life, (3) search for meaning, (4) gratitude, and (5) awe. A path model like this might allow us to further explore whether people are more likely to turn to religious attendance, prayer, or reading when faced with stress. We would also have a better idea of which type of these three religious practices was more likely to foster each of the positive outcomes (e.g., meaning in life) while controlling for all other correlations in the model.

Response: We appreciate this suggestion by Reviewer #1. We find the proposed ordering of the variables and use of a path model intriguing. We did, however, have some reservations about this type of test considering the nature of change/difference scores for the religious behavior variables. As a reminder, these change scores are calculated by subtracting religious behavior before COVID-19 from religious behavior after COVID-19. Consider what this means in terms of COVID-19 stress predicting these variables. Our MANOVA results show that those who decreased and increased in religious devotion are those that reported higher stress from COVID-19. This suggests that it would be challenging to make sense of any of the change variables as linear constructs. An alternative could be to take the absolute value of the change score, but this also presents an issue. While COVID-19 stress may more robustly predict the absolute value change scores, there were also cases in which MANOVA results revealed that increases or decreases in religious devotion trended in opposite directions, such as presence of meaning when the decrease group reported the lowest mean and the increased group reported the highest mean. 

To put forth a good faith effort, however, we conducted the analyses as suggested by Reviewer #1. The reviewer suggested using variables for change in religious service attendance, change in prayer, and change in reading religious literature. We estimated a model with just these three religious behaviors, but also with the recommendation from Reviewer #2 to reinsert all preregistered variables, we also estimated a model that included talking about religion in addition to the other three behaviors. We have posted both models’ data, input, and output on OSF. Consistent with our concerns about the nature of change scores in a path model, we are not recognizing any results that seem theoretically informative, or that would be useful for the literature. We are open, however to additional feedback on what would be an appropriate use of these analyses, but again caution about how to properly create and interpret change scores in a path model context.

Comment: I note here that the “threat to stress to religious change” path summarizes the argument in the introduction. However, the authors include stress as an outcome of religious change in the analyses and results. This highlights one of the many problems with assessing religious change in cross-sectional data (e.g., we can’t know if the people who did not change in devotion had less stress or whether their devotion reduced their stress level). To their credit, the authors do acknowledge this in the Discussion. As another example, with cross-sectional data, we must rely on introspection and self-report as to whether or not devotion actually changed. And people may not be very good at assessing (or reporting) their own level of devotion from Time 1 to Time 2.

Response: We appreciate the acknowledgment of this limitation. It is true that we cannot determine the direction of results from cross-sectional quantitative data. As the reviewer mentioned, we have done our best to acknowledge this in the discussion. We see the quantitative data as providing a fairly intuitive, but informative description of what people reported. We hope, however, that the qualitative aspect of our data provides additional insight into explanation beyond cross-sectional description. We hope the more holistic qualitative data, in addition to some theoretical reasoning, provides insight into potential directionality with a deeper dive into “why” people reported changes. We have added information on pg. __ to try and further highlight this advantage of having both quantitative and qualitative methods.

Comment: Regarding preregistration, I would not agree that this study was preregistered. Although the authors refer the reader to a document listed as “preregistration” on OSF (osf.io/a4jcf/?view_only=a9e3e2b5d79d4ab3b9905e7106784b72), the document was prepared after the data were collected and after the qualitative data were analyzed; the percentages in each of the three groups has changed between the preregistration and manuscript; and there are some variables (e.g., religious coping and prosocial behavior) that are pertinent to the argument in the introduction of the manuscript that are not included in the analyses in the manuscript. Moreover, the pre-registration lists 11-items used to assess COVID stress and perceived threat. However, only five of the items were used for the analyses presented in the manuscript and those five items were separated into two variables. Overall, the “preregistration” and the manuscript analyses are so different that I must conclude the authors are overstating by claiming that this study (at least the quantitative analyses) were pre-registered. This does not preclude publication . . . I think there are real benefits to conducting exploratory research and exploratory research is suitable for publication. But the quantitative analyses were not preregistered, strictly speaking.

Response: We understand the reviewer’s query surrounding our preregistration. We have tried to be as transparent as possible through our decision-making process and hope we can provide some clarification. Data were collected between May 27-31, 2020. We did not test anything qualitatively or quantitatively before an initial preregistration. After cleaning the data and running basic frequencies, we uploaded a preregistration detailing our research question and our qualitative data analysis plan. Once we completed the qualitative data coding and organizing of themes, we used a combination of theory and the information gleaned from the qualitative data to preregister our quantitative analysis.

After we completed our quantitative analyses, we began to realize how unwieldy the manuscript was becoming with so many variables and both quantitative and qualitative analyses. For that reason, we decided to move a number of variables to a supplement, so we could maintain transparency about everything we had tested and simplify the manuscript to focus on some of the big takeaways. Despite preregistering the qualitative analyses before the quantitative analyses, for the flow of the paper, we also began to realize that the qualitative analyses seemed to work better reported after the quantitative analyses. This is because the qualitative analyses seemed to provide a deeper dive into potential explanations for the descriptive picture from the quantitative results.

Consistent with Reviewer 2’s suggestion, we have reinserted all the variables that were initially preregistered, which hopefully adds clarity to our preregistration. On pg. 11 we have also added several sentences clarifying this preregistration process. We are happy to adjust any wording the editor our reviewer requests surrounding our preregistration, as we realize that this step-by-step approach after data are collected is not conventional.

Hopefully this reinsertion also clarifies the concern about items for COVID-19 attitudes and behaviors. We ended up utilizing the 11 items on stress and perceived threat of COVID-19 in addition to two items focused on beliefs about COVID-19. That factor analysis is displayed in the supplemental material and shows how those items were eventually broken down into several subscales. We have struggled finding best terms to describe a number of these COVID-19 variables, as we created the variables toward the beginning of the pandemic in a highly exploratory fashion. If the editor or reviewer has any suggestions to further clarify their use we are happy to incorporate them.

Comment: Finally, I want to add that the manuscript is very well written. However, references #2 through #6 (Geertz, Fromm, Freud, Buber, Durkheim) seem oddly out of place, inappropriate, and irrelevant in this paper. There are plenty of psychology papers that could have been cited to support the argument that religion helps people cope with life’s challenges. (And there are plenty of other papers not cited showing that religion can actually lead to negative coping and spiritual struggles!) Additionally, in terms of references cited, there are only three articles cited focusing specifically on the links between COVID and religion. However, there have been many articles published either supporting, elaborating, or contradicting the findings of this study that are not referenced. I have listed just a few of the 192 articles listed on PsycInfo at the end of this review. Although it is not necessary to cite all of the ones listed below, the authors should be familiar with and at least cite the most relevant.

Additional articles of interest:

Davis, E. B., McElroy-Heltzel, S., Lemke, A. W., Cowden, R. G., VanderWeele, T. J., Worthington, E. L., Jr., . . . Aten, J. D. (2021). Psychological and spiritual outcomes during the COVID-19 pandemic: A prospective longitudinal study of adults with chronic disease. Health Psychology, 40(6), 347-356.

DeRossett, T., LaVoie, D. J., & Brooks, D. (2021). Religious coping amidst a pandemic: Impact on COVID-19-related anxiety. Journal of Religion and Health, 60(5), 3161-3176.

Johnson, K. A., Baraldi, A. N., Moon, J. W., Okun, M. A., & Cohen, A. B. (2021). Faith and science mindsets as predictors of COVID-19 concern: A three-wave longitudinal study. Journal of Experimental Social Psychology, 96, 14.

Kasapo?lu, F. (2022). The relationship among spirituality, self-efficacy, covid-19 anxiety, and hopelessness during the covid-19 process in turkey: A path analysis. Journal of Religion and Health, doi:http://dx.doi.org/10.1007/s10943-021-01472-7

Kranz, D., Niepel, C., Botes, E., & Greiff, S. (2020). Religiosity predicts unreasonable coping with COVID-19. Psychology of Religion and Spirituality, doi:http://dx.doi.org/10.1037/rel0000395

Michaels, J. L., Hao, F., Ritenour, N., & Aguilar, N. (2022). Belongingness is a mediating factor between religious service attendance and reduced psychological distress during the covid-19 pandemic. Journal of Religion and Health, doi:http://dx.doi.org/10.1007/s10943-021-01482-5

Papazoglou, A. S., Moysidis, D. V., Tsagkaris, C., Dorosh, M., Karagiannidis, E., & Mazin, R. (2021). Spiritual health and the COVID-19 pandemic: Impacts on orthodox christianity devotion practices, rituals, and religious pilgrimages. Journal of Religion and Health, 60(5), 3217-3229.

Pirutinsky, S., Cherniak, A. D., & Rosmarin, D. H. (2020). COVID-19, mental health, and religious coping among american orthodox jews. Journal of Religion and Health, 59(5), 2288-2301.

Rigoli, F. (2021). The link between covid-19, anxiety, and religious beliefs in the united states and the united kingdom. Journal of Religion and Health, doi:http://dx.doi.org/10.1007/s10943-021-01296-5

Zacher, H., & Rudolph, C. W. (2021). Individual differences and changes in subjective wellbeing during the early stages of the COVID-19 pandemic. American Psychologist, 76(1), 50-62.

Zhang, H., Hook, J. N., Hodge, A. S., Coomes, S. P., Davis, C. W., Van Tongeren, D. R., . . . Aten, J. D. (2021). Religious and spiritual struggles and coping amidst the COVID-19 pandemic: A qualitative study. Spirituality in Clinical Practice, 8(4), 245-261.

Response: We appreciate the detailed list of references! This was incredibly useful in improving the literature review. We eliminated the references the reviewer recommended, and have incorporated all of the articles suggested where applicable.

Comment: I hope my comments can be helpful to the editor and authors.

Response: We appreciate the comments! We believe the manuscript is undoubtedly improved due to the extensive feedback provided by this reviewer. 

Reviewer #2 

Comment: The manuscript describes a correlational study conducted at the beginning of the Covid-19 pandemic on the effects of the pandemic on change in religious devotion. The results showed rather a trivial finding that some participants increased, some decreased in their religious devotion while others remained unchanged. A more nuanced look into the change in religiosity revealed that while an increase in personal prayer and engagement with the scripture was characteristic for the group with increased religiosity, the opposite was true for the group with decreased religiosity. A qualitative analysis further showed that increased religiosity was associated with increased stress and search for meaning in life while decreased religiosity with less time that could be possibly dedicated to religious activities. Overall, this is a straightforward study with several strengths – pre-registered hypotheses, data coming from a crucial period during the pandemic, and a combination of qualitative and quantitative approaches. As I noted, the findings from this study are not overwhelming, but I find that there is merit to them, especially if the authors would be willing to put more work into the manuscript.

Response: We appreciate that this reviewer recognizes several strengths in our study. We also appreciate the careful critiques and hope that our revisions help address some of the concerns.

Comment: I found the intro a bit lengthy and repetitive. I believe that the writing could be more concise; there is a lot of repetition in the paper.

Response: We appreciate this comment as it allowed us to recognize some redundancy in introducing the topic. We have trimmed some of the material to arrive at the specific research focus more quickly (e.g., the introduction to the introduction is now two pages rather than three). Note that with the suggestion that we reinsert all the preregistered variables, the introduction is now longer overall. However, we have done our best to make these introductions as brief as possible in order to prevent the overall introduction from becoming unwieldy. If the editor or the reviewer see other clear examples of ways to lessen the length, we will be happy to do so.

Comment: Notably, shortening the current manuscript will also allow including information that was moved to the supplementary material but should be included in the main text—namely, the theoretical rationale for all the pre-registered hypotheses and their results. When I was reading the introduction, I was surprised by the inclusion of the gratitude and awe measures that did not seem to follow from any theory. I immediately started to wonder how many measures were obtained and whether including these two measures in the introduction is just because these measures came out as statistically significant in the results section. Indeed, this is what the authors confirmed in the first paragraph on p. 15 where they noted the use of other measures that were moved to supplements because their results were not statistically significant. While I understand the reasons for this decision, I believe it harms the paper because the results feel cherry-picked, and the underlying theory is insufficient. Indeed, after reading the paper, I kept wondering why the authors did not try to address the most interesting question, that is, what predicts an increase or decrease in religiosity due to the Covid-19 pandemic? Looking into the pre-registration file revealed that the authors included several variables to this end. Thus, the introduction should state the theoretical rationale for including these variables and hypotheses, and the results should be reported in the main text. Likewise, the authors should discuss what does the lack of statistically significant results mean for the theory.

Response: We appreciate this reviewer’s perspective and have decided to reinsert all of the preregistered variables into the main text of the paper that we had originally relegated to a supplement. As we mentioned in our response to Reviewer #1, after we initially completed our quantitative analyses, we began to realize how unwieldy the manuscript was becoming with so many variables and both quantitative and qualitative analyses. For that reason, we decided to move a number of variables to a supplement, so we could maintain transparency about everything we had tested, but simplify the manuscript to focus on some of the big takeaways. However, we can see that with succinct introductions to all variables assessed and hypotheses tested, we can maintain a tight focus on biggest takeaways from the manuscript while also providing easy access to all results in the main manuscript.

Comment: Looking further into the dataset the authors provided (thank you for sharing the data), I see that they collected many variables that could be used in an explorative analysis. For instance, it could be assessed whether the number of kids contributed to the decrease in religiosity (as people have less time to participate in religious activities). I will leave this up to the authors, but it seems like the data actually have more potential than discussed in the manuscript. I also feel like the sheer extent of the dataset deserves a comment in the main text.

Response: We appreciate the compliment about the data! On pg. 11, we added information discussing our process in preregistering our variable selection and analysis plan. We have mentioned that not all variables in the dataset were included, and that both the codebook and data are available online for interested readers who might like to dive more deeply into variables or analyses.

Because we have elected to include everything from our preregistration, in the attempt to be as true to what we preregistered as possible, for now we have not added variables to our analyses that we initially preregistered. If, however, the editor or the reviewers feel strongly about any specific variables that should be tested, we are happy to consider including additional variables. Of note, the number of kids was a sociodemographic variable we originally preregistered and is now included in the manuscript.

Comment: Would it be possible to include also the original questionnaire in the OSF repository? I aimed to check some of the analyses conducted in the paper yet could not make sense of the dataset.

Response: We have added the original questionnaire to the OSF repository. We hope this makes it easier to navigate the dataset.

Comment: The reason I wanted to conduct more analyses myself was that I am worried about the correlational nature of the study and whether all major confounds were appropriately taken into account. Generally, MANOVA does not strike as the best analytical approach in this case since there was no random assignment to the experimental groups, and one cannot assume that the groups would be more or less homogenous due to the random assignment. The authors deal with this issue on p. 14 by conducting MANOVAs where increase/decrease of religiosity predicts sociodemographic variables, but this is not sufficient. A regression approach would be more appropriate in this situation since all the effects of covariates could be used to adjust the main investigated effects of religious increase/decrease on the outcome variables of interest. Likewise, the authors do not report any fit indices of their MANOVA models, but I wonder how the normality assumption held with dependent variables being often measured at Likert scales.

Response: This comment has given us the chance to carefully reflect on the appropriateness of our analyses and consider what alternatives might be viable. The most straightforward point to address is the assumption of normality for dependent variables using Likert scales. After consulting various sources, we determined that in cases where the normality assumption is violated (as assessed through Levene’s Test) that it would make sense to follow-up with nonparametric tests. We have reported these tests in all cases that Levene’s Test was significant (see page 21, 24, and 26).

We carefully considered the point about utilizing regression analyses rather than MANOVA models. We recognize that in many ways regression is the preferred analysis for this type of data; as we considered the implications, however, we had a number of reservations we hope to explain.

Our first concern was in potentially having too much confidence in directional propositions. With MANOVA analyses, we have been careful to try and use language surrounding group differences when discussing our analyses, rather than language surrounding prediction, as our analyses cannot determine the direction of association. We have tried to carefully caveat, for example, that we cannot determine for those who reported an increase in religious devotion, and also reported higher gratitude, whether those with higher gratitude are more likely to interpret things in a way to increase their religious devotion or gratitude is something of a byproduct of increasing religious devotion. In our discussion, we theoretically address which may seem more probable directionally, but we stop short of making an assertion. One concern asserting a direction of prediction is that for some variables it seems more theoretically plausible that a construct could be the result of increasing religious devotion (e.g., increased meaning of life) while in other cases it seems more theoretically plausible that religious devotion could be the result of another variable (e.g., stress from COVID-19). With the nature of our exploratory analyses, we have made our best effort to clarify that we are trying to understand basic descriptive differences between three different groups, and we are concerned that some results may become more challenging to interpret if we are forced to make change in religious devotion exclusively an outcome, or exclusively a predictor.

In addition to this theoretical concern, there are also some practical concerns we had about trying to set up regression analyses with change in religious devotion as exclusively a predictor or outcome. If religious change was set up as a categorical outcome of logistic regression, we were concerned about the multicollinearity of so many correlated predictors simultaneously predicting group membership. Particularly with reinserting variables from the supplement, this would add up to 37 simultaneous predictors. With shared variance being controlled for between so many variables, it seems like it would be challenging to make sense of results that emerge.

Alternatively, we struggled to make sense of religious change exclusively as a predictor rather than an outcome. We do not believe it would make sense for it to function as a linear predictor, as there are cases of participants reporting higher scores in the decrease religious devotion group and the increase religious devotion group than the no change group (e.g., stress from COVID-19, searching for meaning), suggesting that they seem to function better as distinct groups rather than a linear predictor. Alternatively, perhaps the three groups could be modeled as several dichotomized categories through dummy variables. But this may still create some problems as this would involve multiple dummy variables potentially being highly correlated enough to create some multicollinearity concerns.

Additionally, as a reminder, these analyses are preregistered and we were concerned about deviating too heavily from analyses we specifically committed to conduct.

Finally, as the manuscript is currently structured, the three groups in the quantitative analysis map cleanly onto the three groups in the qualitative coding. This makes it easier to synthesize results from both types of analysis. Furthermore, we believe the completely distinct patterns of coded responses in the qualitative data highlight that the three groups can be understood categorically, as there was almost no overlap in the types of codes in their responses. 

If we are misunderstanding the reviewer’s suggestion and there is a clear path forward with regression analyses that does not incur the costs we have highlighted, we would be glad to do so. For now, we have focused on the nonparametric analyses as something of a robustness test, but we are open to additional discussion on the best path forward on this matter.

Concerning control variables, as a reminder, we were initially planning on conducting the analyses with sociodemographic variables as controls if there were significant differences across change in religious devotion. Since we did not, however, find any differences across groups, we did not end up with any sociodemographic controls.

Comment: I believe that the work of Bronislaw Malinowski and his contemporary followers should be discussed in the manuscript since it could help explain some of the findings regarding the effects of stress and uncertainty. For example, see the two references below:

Lang, M., Krátký, J., & Xygalatas, D. (2020). The role of ritual behavior in anxiety reduction: An investigation of Marathi religious practices in Mauritius. Philosophical Transactions of the Royal Society B, 375(1805), 20190431. https://doi.org/https://doi.org/10.1098/rstb.2019.0431

Sosis, R., & Handwerker, W. P. (2011). Psalms and coping with uncertainty: Religious Israeli women’s responses to the 2006 Lebanon war. American Anthropologist, 113(1), 40–55. https://doi.org/10.1111/j.1548-1433.2010.01305.x

Response: We appreciate being made aware of these references and we have incorporated them into the manuscript.

Comment: The fact that pre-registration was conducted after data collection should be noted in the paper (apologies if I have missed it)

Response: On pg. 11, in our more detailed explanation of our process in preregistering and analyzing the data, we have explicitly added that analyses were preregistered after data collection.

Comment: Overall, I believe that after working on the manuscript a bit more, it would be a valuable contribution to the current literature on the relationship between religion and stressful life events.

Response: We appreciate the compliment and valuable feedback! We hope the changes we have made to the manuscript address the points mentioned and we remain open to additional feedback in further strengthening the manuscript.

---

## [Decision Letter · Decision Letter 1]

3 Nov 2022

PONE-D-22-03669R1Turning Toward or Away from God: COVID-19 and Changes in Religious DevotionPLOS ONE

Dear Dr. Leonhardt,

Thank you for submitting your manuscript to PLOS ONE. After careful consideration, we feel that it has merit but does not fully meet PLOS ONE’s publication criteria as it currently stands. Therefore, we invite you to submit a revised version of the manuscript that addresses the points raised during the review process.

We look forward to receiving your revised manuscript.

Kind regards,

Andrea Fronzetti Colladon, Ph.D.

Academic Editor

PLOS ONE

Journal Requirements:

Reviewers' comments:

Reviewer's Responses to Questions

**Comments to the Author**

1. If the authors have adequately addressed your comments raised in a previous round of review and you feel that this manuscript is now acceptable for publication, you may indicate that here to bypass the “Comments to the Author” section, enter your conflict of interest statement in the “Confidential to Editor” section, and submit your "Accept" recommendation.

Reviewer #2: (No Response)

2. Is the manuscript technically sound, and do the data support the conclusions?

Reviewer #2: Partly

3. Has the statistical analysis been performed appropriately and rigorously? 

Reviewer #2: N/A

4. Have the authors made all data underlying the findings in their manuscript fully available?

Reviewer #2: Yes

5. Is the manuscript presented in an intelligible fashion and written in standard English?

Reviewer #2: Yes

6. Review Comments to the Author

Reviewer #2: I would like to thank the authors for considering my suggestions and for updating the manuscript. The revised version introduced some improvements and is now more transparent about the entire research process. This is an important change because such transparency will allow readers to judge what can be inferred from this study (given the relatively large number of variables tested). After reading the revised version, I stand with my previous assessment of this study – the findings are relatively trivial (showing variables correlating with increase and/or decrease in religiosity), and the most interesting question posed by the authors remains unanswered: that is, why some people react to stress by an increase in religiosity, some do nothing, and some with decrease? The importance of this question is illustrated by the positive association of an increase in religiosity with variables indexing well-being and prosocial emotions. Hence, not properly addressing this question feels a bit frustrating for me as a reader. To be fair, the authors provide a hypothetical answer to this question relying on internal and external religiosity (p. 34). However, this answer is not supported by the presented data because these variables were not assessed.

From the sheer number of results the authors provide, a different (speculative) explanation seems probable. Namely, that people who were already susceptible to a decline in the intensity of their faith did so due to Covid-related stress and demands. Looking at the first block of results in Table 1 (Sociodemographic), no statistically significant differences were reported. However, if we look at the overall profile of people who decreased in their religiosity, we see that they have, on average, higher education, fewer kids, higher income, belong to a higher social class, and are liberal. In one way or another, all these variables have been implied in the increasing secularization of Western countries. Why these variables were not significant may have to do with the chosen statistical test.

In response to my argument that the data should be analyzed using regressions, the authors made several statements that do not seem entirely correct to me. Let me explain point by point.

The authors wrote: “Our first concern was in potentially having too much confidence in directional propositions. With MANOVA analyses, we have been careful to try and use language surrounding group differences when discussing our analyses, rather than language surrounding prediction, as our analyses cannot determine the direction of association.” I agree, but the same applies to regression coefficients, which are identical to MANOVA if the regression equation is set up with the same parameters as MANOVA! A simple regression coefficient is nothing else than a correlational coefficient. Thus, adjusting these estimates for variables that could potentially confound the assumed causal link is even more important.

Further, the authors argued: “we are concerned that some results may become more challenging to interpret if weare forced to make change in religious devotion exclusively an outcome, or exclusively a predictor.” Again, the regression equation is set as if some variables are predictors and some outcomes, but it’s still the same result even if those variables are switched (and the same will be true for MANOVA!). Hence, this is not about the test being used but about the assumptions that should be explicitly stated.

Next, the authors wrote: “were concerned about the multicollinearity of so many correlated predictors simultaneously predicting group membership. Particularly with reinserting variables from the supplement, this would add up to 37 simultaneous predictors.” Good point, yet it begs the question why the authors measured 37 predictors? I would be personally happy to see just a model where the probability of belonging either to a decrease or increase category would be predicted by an interaction between Covid-related stress and some of the sociodemographic variables (education, social class, political affiliation). However, I concur with Reviewer #1 that a path model would be more revealing.

Next point: “Alternatively, perhaps the three groups could be modeled as several dichotomized categories through dummy variables. But this may still create some problems as this would involve multiple dummy variables potentially being highly correlated enough to create some multicollinearity concerns.” Again, this is exactly what MANOVA does. Creating dummy variables based on the change in religiosity will produce the same results in linear regression as in MANOVA.

Finally, “these analyses are preregistered and we were concerned about deviating too heavily from analyses we specifically committed to conduct.” This is a valid point. In my view, however, the point of pre-registration is to be transparent about the research process. What is better – rigidly sticking to an inferior analytical approach or explaining the change to a more informative approach?

Ultimately, the decision about which statistical test to use is up to the authors. All have their pros and cons. In my replies, I just wanted to point out that the MANOVA results can be reproduced using linear regression and that the regression approach offers further flexibility to make the models more informative.

Minor:

Thank you for sharing your questionnaire. It did not help me to understand the data set, though. The order of variables does not seem to follow the order of questions, and I was still confused about what variable indexes which question. A proper codebook explaining each variable and its range would be necessary.

7. PLOS authors have the option to publish the peer review history of their article (what does this mean?). If published, this will include your full peer review and any attached files.

Reviewer #2: No

---

## [Author Response · Author response to Decision Letter 1]

30 Dec 2022

Andrea Fronzetti Colladon, Ph.D.

Academic Editor

PLOS ONE

Dr. Colladon,

We appreciate the thoughtful comments throughout this revision process. We believe we have been able to address the last comments from Reviewer 2 for this “Minor Revision” decision. If there is anything else that we can do to clarify points or address concerns, we are happy to do so.

Reviewer 2’s Comments

Comment: I would like to thank the authors for considering my suggestions and for updating the manuscript. The revised version introduced some improvements and is now more transparent about the entire research process. This is an important change because such transparency will allow readers to judge what can be inferred from this study (given the relatively large number of variables tested). 

Response: We appreciate the compliment!

Comment: After reading the revised version I stand with my previous assessment of this study – the findings are relatively trivial (showing variables correlating with increase and/or decrease in religiosity), and the most interesting question posed by the authors remains unanswered: that is, why some people react to stress by an increase in religiosity, some do nothing, and some with decrease? The importance of this question is illustrated by the positive association of an increase in religiosity with variables indexing well-being and prosocial emotions. Hence, not properly addressing this question feels a bit frustrating for me as a reader. To be fair, the authors provide a hypothetical answer to this question relying on internal and external religiosity (p. 34). However, this answer is not supported by the presented data because these variables were not assessed.

Response: We share this reviewer’s frustrations regarding the limitations of the data collected in this study. Particularly with cross-sectional data there are challenges to be fully confident in some of the potential explanations we suggest. We have done our best to acknowledge this in the limitations section. We do, however, think that the strength of combining quantitative and qualitative data help us to make some tentative assertions. We appreciate the reviewer raising the point specifically about internal versus external religiosity as it helped us realize that we can more clearly point to the qualitative evidence in support of our assertions. On pg. 33-34, we have added the following statement to more clearly highlight some additional evidence on this point of internal and external religiosity:

“Our qualitative data, in particular, seem to speak to this distinction between focusing on more internal versus external religiosity. Among those who reported an increase in religious devotion, 47.7% participants focused their comments specifically on engaging in personal worship such as prayer or scripture study; alternatively, among those who reported a decrease in religious devotion, 68.9% mentioned the lack of engagement in communal worship. This difference is striking given that we did not specifically prompt participants to report on these experiences denoting internal or external religiosity. Rather they spontaneously reported them in response to a broad, open-ended question about “Why” their religious devotion had changed or not changed since COVID-19.”

While we acknowledge our inability to state with complete confidence what factors drive the difference between the increase in religious devotion and decrease in religious devotion groups, we hope that this reminder of additional qualitative evidence at least adds to the credibility of our assertions about this internal versus external focus being part of the picture.

Comment: From the sheer number of results the authors provide, a different (speculative) explanation seems probable. Namely, that people who were already susceptible to a decline in the intensity of their faith did so due to Covid-related stress and demands. Looking at the first block of results in Table 1 (Sociodemographic), no statistically significant differences were reported. However, if we look at the overall profile of people who decreased in their religiosity, we see that they have, on average, higher education, fewer kids, higher income, belong to a higher social class, and are liberal. In one way or another, all these variables have been implied in the increasing secularization of Western countries. Why these variables were not significant may have to do with the chosen statistical test… 

I would be personally happy to see just a model where the probability of belonging either to a decrease or increase category would be predicted by an interaction between Covid-related stress and some of the sociodemographic variables (education, social class, political affiliation). However, I concur with Reviewer #1 that a path model would be more revealing.

Response: This is an intriguing theory. As requested, we conducted analyses with Covid-related stress interacting with education, social class, and political affiliation, predicting the likelihood of increasing versus decreasing in religious devotion (dichotomized, as we believe this best fits the description of what the reviewer is suggesting). We conducted the analyses with all predictors simultaneously, as well as separate models of each predictor interacting with Covid-related stress. All analyses were conducted with path models, consistent with the suggestion of the reviewer. We did not find any significant results. If there are further analyses on this point that are desired, we are open to them. We have posted the analyses on OSF. 

Just because we did not find anything does not negate the value in this point. We still believe this is worth exploring more deeply in other data. We have added information to the discussion section (pg. 37-38) highlighting that it would be valuable to assess this more closely in future research.

“We note that although there were not significant differences between groups for sociodemographic variables, that the means for those in the decrease group seemed to suggest higher education, fewer kids, higher income, belonging to a higher social class, and being liberal. As these variables are connected to secularization of Western countries, it could be valuable for future research to more closely explore whether those who are already susceptible to a decline in the intensity of their faith may have done so due to COVID-19 related stress and demands.”

Comment: In response to my argument that the data should be analyzed using regressions, the authors made several statements that do not seem entirely correct to me. Let me explain point by point.

The authors wrote: “Our first concern was in potentially having too much confidence in directional propositions. With MANOVA analyses, we have been careful to try and use language surrounding group differences when discussing our analyses, rather than language surrounding prediction, as our analyses cannot determine the direction of association.” I agree, but the same applies to regression coefficients, which are identical to MANOVA if the regression equation is set up with the same parameters as MANOVA! A simple regression coefficient is nothing else than a correlational coefficient. Thus, adjusting these estimates for variables that could potentially confound the assumed causal link is even more important.

Further, the authors argued: “we are concerned that some results may become more challenging to interpret if we are forced to make change in religious devotion exclusively an outcome, or exclusively a predictor.” Again, the regression equation is set as if some variables are predictors and some outcomes, but it’s still the same result even if those variables are switched (and the same will be true for MANOVA!). Hence, this is not about the test being used but about the assumptions that should be explicitly stated.

Next, the authors wrote: “were concerned about the multicollinearity of so many correlated predictors simultaneously predicting group membership. Particularly with reinserting variables from the supplement, this would add up to 37 simultaneous predictors.” Good point, yet it begs the question why the authors measured 37 predictors? …

Next point: “Alternatively, perhaps the three groups could be modeled as several dichotomized categories through dummy variables. But this may still create some problems as this would involve multiple dummy variables potentially being highly correlated enough to create some multicollinearity concerns.” Again, this is exactly what MANOVA does. Creating dummy variables based on the change in religiosity will produce the same results in linear regression as in MANOVA.

Finally, “these analyses are preregistered and we were concerned about deviating too heavily from analyses we specifically committed to conduct.” This is a valid point. In my view, however, the point of pre-registration is to be transparent about the research process. What is better – rigidly sticking to an inferior analytical approach or explaining the change to a more informative approach?

Ultimately, the decision about which statistical test to use is up to the authors. All have their pros and cons. In my replies, I just wanted to point out that the MANOVA results can be reproduced using linear regression and that the regression approach offers further flexibility to make the models more informative.

Response: The reviewer has brought up several fair points. We believe that a regression approach could be justified. We appreciate, however, the acknowledgment that the statistical test is ultimately up to us as authors. With this acknowledgment, we maintain that the MANOVA approach is still defensible and best suits the purposes of this particular paper.

A primary concern with switching to a regression approach concerns the fact that the language of prediction inherent in the use of this approach often insinuates directionality, which of course we cannot determine from our cross-sectional data. There is room for disagreement here, as the reviewer correctly makes a fair point about regressions really not being more than correlation coefficients. But we believe it would be more challenging to navigate this particular manuscript with language of prediction than language of group differences from MANOVA. Additionally, we believe the results become more digestible and interpretable with such clear cut groups and group difference language, especially since they are they same groups used for the qualitative coding. 

Another concern with a potential regression approach is that we do have 37 variables to assess; we appreciate the reviewer’s agreement that this many variables in a regression would likely create problems with multicollinearity. It is true that 37 variables is a lot. In writing the manuscript, we began to realize how things could potentially become unwieldy (which is why in our original submission we placed many of the results in a supplemental file). We did, however, accept the feedback of the reviewer and editor in reinserting all of the variables that were initially preregistered. Though it is a lot of variables, we did our best to ensure there were at least theoretically grounded reasons for why these variables may be valuable to include. We maintain, consistent with our preregistration that all the variables have at least some theoretical basis for how they might explain group differences between those who increase, stay the same, or decrease in religiosity. With the exploratory nature of the study it made sense to account for them. With all of them needing accounting, we maintain that a MANOVA is beneficial in avoiding potential multicollinearity concerns that would likely emerge from a regression approach.

Finally, at the end of the day, we ultimately did preregister MANOVA analyses. An argument could be made for alternative analyses. Considering, however, there is not necessarily anything incorrect in the use of MANOVA, and the acknowledgment from the reviewer that the decision is up to us as authors, we prefer to stay consistent with what we preregistered.

Comment: Thank you for sharing your questionnaire. It did not help me to understand the data set, though. The order of variables does not seem to follow the order of questions, and I was still confused about what variable indexes which question. A proper codebook explaining each variable and its range would be necessary.

Response: We apologize for the lack of clarity for some aspects of the codebook. We have ensured that all variables are presented in the same order in the codebook and dataset. We also have added all the stems in the codebook that are used to label variables in the dataset (e.g., the stem of “awe” being in parentheses next to the awe measure in the codebook so that interested readers can more clearly see that the six items for awe in the dataset have the stem of “awe”, such as awe1, awe2, etc.). We also have made sure that the range is shown in the codebook for all quantitative measures.

---

## [Editor Report · Decision Letter 2]

10 Jan 2023

Turning Toward or Away from God: COVID-19 and Changes in Religious Devotion

PONE-D-22-03669R2

Dear Dr. Leonhardt,

We’re pleased to inform you that your manuscript has been judged scientifically suitable for publication and will be formally accepted for publication once it meets all outstanding technical requirements.

Kind regards,

Andrea Fronzetti Colladon, Ph.D.

Academic Editor

PLOS ONE

---

## [Editor Report · Acceptance letter]

27 Feb 2023

PONE-D-22-03669R2 

Turning toward or away from God: COVID-19 and changes in religious devotion 

Dear Dr. Leonhardt:

I'm pleased to inform you that your manuscript has been deemed suitable for publication in PLOS ONE. Congratulations! Your manuscript is now with our production department. 

Kind regards, 

on behalf of

Prof. Andrea Fronzetti Colladon 

Academic Editor

PLOS ONE